# Endoplasmic Reticulum Stress Cooperates in Silica Nanoparticles-Induced Macrophage Apoptosis via Activation of CHOP-Mediated Apoptotic Signaling Pathway

**DOI:** 10.3390/ijms20235846

**Published:** 2019-11-21

**Authors:** Fenglei Chen, Jiaqi Jin, Jiahui Hu, Yujing Wang, Zhiyu Ma, Jinlong Zhang

**Affiliations:** 1College of Veterinary Medicine, Yangzhou University, Yangzhou 225009, Jiangsu, China; savannahkini@163.com (J.J.); Alohahujiahui@163.com (J.H.); Pluto7107@163.com (Y.W.); mzy2017@yzu.edu.cn (Z.M.); zjl@yzu.edu.cn (J.Z.); 2Jiangsu Co-innovation Center for Prevention and Control of Important Animal Infectious Diseases and Zoonoses, Yangzhou 225009, Jiangsu, China

**Keywords:** silica nanoparticles, endoplasmic reticulum stress, apoptosis, CHOP

## Abstract

While silica nanoparticles (SiNPs) have wide applications, they inevitably increase atmospheric particulate matter and human exposure to this nanomaterial. Numerous studies have focused on how to disclose SiNP toxicity and on understanding its toxic mechanisms. However, there are few studies in the literature reporting the interaction between endoplasmic reticulum (ER) stress and SiNP exposure, and the corresponding detailed mechanisms have not been clearly determined. In this study, CCK-8 and flow cytometry assays demonstrated that SiNPs gradually decreased cell viability and increased cell apoptosis in RAW 264.7 macrophage cells in dose- and time-dependent manners. Western blot analysis showed that SiNPs significantly activated ER stress by upregulating GRP78, CHOP, and ERO1α expression. Meanwhile, western blot analysis also showed that SiNPs activated the mitochondrial-mediated apoptotic signaling pathway by upregulating BAD and Caspase-3, and downregulating the BCL-2/BAX ratio. Moreover, 4-phenylbutyrate (4-PBA), an ER stress inhibitor, significantly decreased GRP78, CHOP, and ERO1α expression, and inhibited cell apoptosis in RAW 264.7 macrophage cells. Furthermore, overexpression of CHOP significantly enhanced cell apoptosis, while knockdown of CHOP significantly protected RAW 264.7 macrophage cells from apoptosis induced by SiNPs. We found that the CHOP-ERO1α-caspase-dependent apoptotic signaling pathway was activated by upregulating the downstream target protein ERO1α and caspase-dependent mitochondrial-mediated apoptotic signaling pathway by upregulating Caspase-3 and downregulating the ratio of BCL-2/BAX. In summary, ER stress participated in cell apoptosis induced by SiNPs and CHOP regulated SiNP-induced cell apoptosis, at least partly, via activation of the CHOP-ERO1α-caspase apoptotic signaling pathway in RAW 264.7 macrophage cells.

## 1. Introduction

Following the rapid development of nanotechnology, the potential harmful health effects of nanomaterial have aroused the public’s widespread attention during the past decades [1,2]. Due to the unique physicochemical properties of nanoparticles (NPs), they can reach various tissues and organs in the whole body, and penetrate the body’s protective barriers, such as the blood brain barrier, to increase the risk of toxicity to humans and animals. Among these NPs, silica nanoparticles (SiNPs) are a popular inorganic material for engineered nanoparticles. SiNPs have been extensively developed for chemical, materials, cosmetics, food, and biomedical applications, such as disease diagnosis, drug delivery, cancer therapy, and new types of animal vaccine preparations [3,4,5]. The wide application of SiNPs inevitably increases air pollutants. A previous study found that Si is a component with one of the highest contents in ambient fine particulate matter (PM 2.5) [6]. With the larger specific surface area and the higher particle number, SiNPs can adsorb much higher concentrations of pollutants and aggravate air pollution, and be easily exposed to humans through inhalation, ingestion, skin contact, and even vein injection [7,8,9]. SiNPs are currently on the priority lists for toxicological evaluation by the Organization for Economic Cooperation and Development (OECD) [10]. It is a prerequisite for the application of SiNPs to fully disclose its toxicity and deeply understand its toxic mechanisms.

In recent years, the toxicological studies of SiNPs showed that one of the most common toxic mechanisms is that SiNPs can produce excessive reactive oxygen species (ROS) in the body. Previous studies demonstrated that SiNPs induced cytotoxicity and oxidative stress through glutathione depletion and ROS generation in human bronchoalveolar carcinoma-derived cells (A549), human bronchial epithelial cells (Beas-2B), RAW 264.7 macrophage cells, myocardial H9c2(2-1) cells, renal proximal tubular cell lines (human HK-2 and porcine LLC-PK1), and human keratinocyte cells (HaCaT) [11,12,13,14,15,16,17]. Further studies demonstrated that SiNPs induced hepatocellular carcinoma cell line (HepG2) apoptosis through mitochondrial pathway mediated oxidative stress due to excessive ROS production [18,19,20]. Other than oxidative stress, excessive ROS affects endoplasmic reticulum (ER) homeostasis leading to ER stress. ER stress is induced by various pathophysiological conditions, such as the production of excessive ROS, the accumulation of misfolded or unfolded proteins, and the imbalance of intracellular Ca^2+^. To restore ER function, ER stress activates molecular chaperones in the ER, such as glucose-regulated protein 78 (GRP78), CCAAT/enhancer binding protein homologous protein (CHOP), and ER oxidoreduclin 1α (ERO1α). GRP78 shows a marked upregulation in ER chaperones in response to ER stress [21]. However, severe or prolonged ER stress will trigger cell apoptosis. CHOP, as an apoptotic transcriptional factor induced by ER stress, is an essential marker for the assessment of ER stress-induced apoptosis [22].

At present, numerous studied have reported that ER stress cooperated in SiNP-induced apoptosis. Christen et al. reported that SiNPs could induce perturbations of the ER, leading to an ER stress response in human hepatoma cells (Huh7) [23]. Yang et al. found that SiNPs caused intracellular Ca^2+^ to increase, leading to intrinsic apoptosis in neuroblastoma SH-SY5Y cells via the CytC/Apaf-1 pathway [24]. ER stress activated SiNPs and oxidized low-density lipoprotein (oxLDL)-induced RAW 264 macrophage cells apoptosis, and the PERK signaling pathway cooperated in this process [25]. ER stress-mitochondria-mediated apoptotic pathways promoted SiNP-induced human umbilical vein endothelial cell (HUVEC) apoptosis, of which CHOP and XBP-1 were upregulated [10]. PERK and ATF6-mediated ER stress was validated to regulate SiNP-induced autophagy via the upregulation of downstream ATF4 and CHOP in hepatocytes [26]. ER stress cooperated in SiNP-induced apoptosis in human pulmonary alveolar epithelial cells (HPAEpiC) [27]. IRE1α-mediated ER stress activated SiNP-induced human endothelial cell (EC) apoptosis [28]. However, only a few literatures reported the interaction between ER stress and SiNP exposure, and the corresponding detailed mechanisms have not been clearly determined. Therefore, in the current research, we studied the effects on ER stress of using RAW 264 macrophage cells in vitro in order to gain insight into the potential toxicity and molecular mechanisms of SiNPs.

## 2. Results

### 2.1. Silica Nanoparticle (SiNP) Characterization and Cell Viability and Apoptosis in RAW 264.7 Macrophage Cells

SiNPs used in this study were near-spherical, porous, and unmodified. The diameter of SiNPs determined by TEM was 5–15 nm (Figure 1A), which was consistent with the size provided by the manufacturer. SiNPs usually displayed good dispersity in solution, although aggregates could be observed. Furthermore, to detect the electrical property of SiNPs, the result showed that SiNPs were electronegative in water, and the zeta potential was measured as −9.9 ± 1.1 mV.

The toxic effects of SiNPs on RAW 264.7 macrophage cells were examined by exposure to 0–200 µg/mL SiNPs for 12 and 24 h. The effects of SiNPs on cell viability were assessed using the CCK-8 assay. SiNPs significantly reduced cell viability of RAW 264.7 macrophage cells at different doses for 12 and 24 h in dose- and time-dependent manners (Appendix A; Figure 1B). The results of flow cytometry analysis revealed that the apoptotic rate of the cells was significantly different when exposed to 0, 50, 100, and 150 µg/mL SiNPs in a dose-dependent manner for 12 and 24 h (Appendix A; Figure 1C,D).

### 2.2. Effect of SiNPs on the Expression of Endoplasmic Reticulum (ER) Stress-Related Proteins in RAW 264.7 Macrophage Cells

To investigate whether ER stress was activated in SiNP-induced apoptosis, the expression of the ER stress-related proteins GRP78, CHOP, and ERO1α were determined in SiNP-exposed RAW 264.7 macrophage cells via western blot analysis. The result showed that SiNPs significantly upregulated the expression of GRP78, CHOP, and ERO1α after exposure to 0, 50, 100, and 150 µg/mL SiNPs for 12 h (Figure 2A,B). We also detected the expression of GRP78, CHOP, and ERO1α at different times (0, 6, 12, and 24 h) after exposure to 100 µg/mL SiNPs. The result also showed that GRP78, CHOP, and ERO1α were upregulated, especially for CHOP and ERO1α in a time-dependent manner (Figure 2C,D).

### 2.3. Effect of SiNPs on the Expression of Apoptosis-Related Proteins in RAW 264.7 Macrophage Cells

To determine whether the mitochondrial apoptotic signaling pathway was also activated, the B-cell lymphoma 2 (BCL-2) family members and Caspase-3 were detected in SiNP-exposed RAW 264.7 macrophage cells via western blot analysis. The result showed that SiNPs significantly upregulated the expression of the proapoptotic protein BCL-2-associated death promoter (BAD) and cleaved Caspase-3, while they downregulated the ratio of BCL-2/BCL-2-associated X protein (BAX) after exposure to 0, 50, 100, and 150 µg/mL SiNPs for 12 h in a dose-dependent manner (Figure 3A,B). We also detected the expression of BCL-2, BAX, BAD, and cleaved Caspase-3 at different times (0, 6, 12 and 24 h) after exposure to 100 µg/mL SiNPs. The result also showed that BAD and cleaved Caspase-3 were upregulated, especially for cleaved Caspase-3 in a time-dependent manner, while the ratio of BCL-2/BAX was downregulated (Figure 3C,D).

### 2.4. Effect of 4-PBA on SiNP-Induced Apoptosis in RAW 264.7 Macrophage Cells

To confirm whether ER stress regulates SiNP-induced cell apoptosis, RAW 264.7 macrophage cells were treated with SiNPs in the presence or absence of the ER stress inhibitor 4-PBA. Similar results were observed using the CCK-8 and flow cytometry assays. The results of CCK-8 showed that pretreatment with 1 mM 4-PBA for 1 h significantly promoted cell viability after exposure to 0, 50, 100, and 150 µg/mL SiNPs for 24 h (Figure 4A). The results of flow cytometry showed that pretreatment with 1 mM 4-PBA significantly inhibited cell apoptosis after exposure to 0, 50, 100, and 150 µg/mL SiNPs for 24 h, respectively (Figure 4B,C). Meanwhile, 4-PBA markedly reduced the immunofluorescence staining of GRP78 (green fluorescence, Alexa Fluor^®^ 488) and CHOP (red fluorescence, Alexa Fluor^®^ 647) after exposure to 100 µg/mL SiNPs for 12 h in RAW 264.7 macrophage cells (Figure 5A; Appendix A). In addition, the results of western blot also revealed that the protein levels of GRP78, CHOP, ERO1α, and cleaved Caspase-3 were significantly decreased in the SiNPs + 4-PBA-exposed RAW 264.7 macrophage cells, while the ratio of BCL-2/BAX was not significantly increased compared with the SiNP exposure group (Figure 5B,C).

### 2.5. Verification of the Overexpression and Knockdown Efficiency of Recombinant CCAAT/Enhancer Binding Protein Homologous (CHOP) Lentivirus Vectors in RAW 264.7 Macrophage Cells

The recombinant lentivirus vectors (Lv)-CHOP and three shCHOPs were successfully constructed. Lentivirus packaging and titration determination were processed in HEK 293T cells. Lv-CHOP, Lv-Control, three shCHOPs, and shNC lentivirus were transduced into RAW 264.7 macrophage cells. The stably expressed cells of CHOP overexpression and knockdown were selected using puromycin (Figure 6A). The transduction efficiency was more than 90% via flow cytometry assays (Appendix A). The results of western blot showed that the expression of CHOP in the Lv-CHOP group was significantly increased compared with that in the Lv-Control group (Figure 6B,C). Meanwhile, the results of immunofluorescence staining also showed that CHOP was primarily located in the nucleus of RAW 264.7 macrophage cells and the fluorescence intensity of CHOP in the Lv-CHOP group was stronger than that in the Lv-Control group (Figure 6D; Appendix A). Three shCHOPs lentiviruses markedly downregulated the expression of CHOP compared with that of the shNC group in the tunicamycin (Tm, 10 µg/mL) and SiNP (100 µg/mL)-exposed RAW 264.7 macrophage cells for 12 h, with shCHOP-1 (50.7% and 44.7%, respectively), shCHOP-2 (78% and 73.5%, respectively), and shCHOP-3 (76.5% and 89.1%, respectively) (Figure 6E–H).

### 2.6. Regulation of CHOP on SiNP-Induced Cell Apoptosis in RAW 264.7 Macrophage Cells

To determine the role of CHOP on SiNP-induced cell apoptosis, the stable cell lines of CHOP overexpression and knockdown were exposed to 0, 50, 100, and 150 µg/mL SiNPs for 24 h. The results of flow cytometry showed that the apoptotic rate of the Lv-CHOP group (26.45%) was higher than that of the Lv-Control group (20.51%) (Figure 7A,B), while the apoptotic rates of the shCHOP-2 (15.75%) and shCHOP-3 groups (14.32%) were lower than that of the shNC group (20.42%) (Figure 7A,B). We further detected the role of the shCHOP-3 group in inhibiting apoptosis after exposure to 200 µg/mL SiNPs for 24 h. The results of flow cytometry showed that the apoptotic rates of the shCHOP-3 group (26.58%) were lower than that of the shNC group (41.20%) (Appendix A). Under the condition of no treatment, there is no significant difference between the Lv-Control group (5.47%) and the Lv-CHOP group (6.19%), while the apoptotic rates of the shCHOP-3 group (3.81%) were lower than that of the Lv-CHOP group (6.19%) (Appendix A).

### 2.7. Effect of CHOP Overexpression and Knockdown on the Expression of ER Stress and Apoptosis-Related Proteins in RAW 264.7 Macrophage Cells

To confirm whether CHOP activated the mitochondrial-mediated apoptotic signaling pathway, we detected the expression of GRP78, ERO1α, BCL-2, BCL-xL, MCL-1, PUMA, BAX, and cleaved Caspase-3. Western blot analysis showed that CHOP overexpression significantly upregulated ERO1α, PUMA, and cleaved Caspase-3, while it downregulated GRP78, BCL-xL, MCL-1, and the ratio of BCL-2/BAX in the Lv-CHOP group (Figure 8A,B; Appendix A). However, CHOP knockdown significantly downregulated ERO1α, PUMA, and cleaved Caspase-3 in the shCHOP groups, especially the shCHOP-2 and shCHOP-3 groups, and upregulated BCL-xL, MCL-1, and the ratio of BCL-2/BAX, while it did not have a significant effect on GRP78 expression in the shCHOP-2 and shCHOP-3 groups (Figure 8A,B; Appendix A).

## 3. Discussion

Previous studies demonstrated that SiNPs caused high toxicity and induced apoptosis in different cell types in vivo and in vitro [29,30,31,32,33,34,35]. However, little is known about the detailed molecular mechanisms. In the current study, we demonstrated that SiNPs induced cell apoptosis, at least partly, via ER stress activation in RAW 264.7 macrophage cells, of which the CHOP-mediated apoptotic signaling pathway played an important role in ER stress-induced apoptosis. Consistent with the previous study [25,36,37], our results revealed that SiNPs significantly inhibited cell viability and induced apoptosis in RAW 264.7 macrophage cells. The induction of ER stress by SiNP exposure was evidenced by detection of the ER stress-related proteins, GRP78, CHOP, and ERO1α. 4-PBA, an ER stress inhibitor, could inhibit SiNP-induced GRP78 and CHOP expression in hepatocytes and HPAEpiC [26,27]. Consistent with these studies, our results showed that 4-PBA decreased GRP78, CHOP, and ERO1α expression, and protected RAW 264.7 macrophage cells from apoptosis induced by SiNP exposure. Furthermore, overexpression of CHOP increased SiNP-induced apoptosis, while knockdown of CHOP decreased SiNP-induced apoptosis. CHOP overexpression enhanced the CHOP-mediated and caspase-dependent mitochondrial-mediated apoptotic signaling pathway by decreasing BCL-xL, MCL-1, and the ratio of BCL-2/BAX and increasing downstream ERO1α, PUMA, and Caspase-3 expression, while CHOP depletion inhibited the apoptotic pathway.

The toxic molecular mechanisms of SiNPs have been studied for decades. One of the mechanisms is that SiNPs produce excessive ROS, which disrupts the balance of Ca^2+^ in the ER [10,24]. The ER is the major store for cellular Ca^2+^ and releasing abundant Ca^2+^ from the ER to the cytoplasm leads to cell death [38]. Depletion of Ca^2+^ stores in the ER disrupts protein folding, processing, and formation, leading to accumulation of unfolded proteins. The disruption of Ca^2+^ homeostasis and accumulation of unfolded proteins in the ER induces ER stress. ER stress initially attenuates protein translation, promotes protein folding, and activates the misfolded/unfolded protein degradation to restore cellular homeostasis [39]. However, with severe or prolonged ER stress, signaling switches from pro-survival to pro-apoptotic. In the present study, after SiNP exposure, GRP78 and CHOP were significantly augmented in RAW 264.7 macrophage cells. These results are consistent with previous studies that SiNPs could increase the expression of GRP78 and CHOP [10,25,27,28]. GRP78 is a typical marker of ER stress activation and induced high expression at the early stage of stress [21]. SiNP exposure increased the expression of GRP78 at the different doses and times, indicating the onset of ER stress in RAW 264.7 macrophage cells after SiNP exposure. CHOP is a key molecule in ER stress-mediated apoptosis, severe and prolonged stress activated CHOP-mediated apoptosis [22]. CHOP upregulation showed that ER stress regulated SiNP-induced apoptosis. Recent studies showed that SiNPs triggered ER stress and then induced apoptosis, of which CHOP increased [10,25,27,28]. The present study further confirmed that SiNPs induced apoptosis, at least partly, via activation of the CHOP-mediated signaling pathway.

As a transcription factor, CHOP does not directly cooperate in cell apoptosis, which interacts with downstream target protein ERO1α to induce cell apoptosis [40,41,42]. ERO1α is an important oxidase and plays an essential role in disulfide bond formation of secreted and cell surface proteins at the posttranslational level [43,44]. In addition to its role in normal cellular functions, ERO1α also cooperates in ER stress-induced apoptosis. As a target of CHOP, the CHOP/ERO1α pathway underlies hepatocellular apoptosis during acute liver failure (ALF) and ER stress-induced apoptosis in macrophages [41]. In the present study, in addition to GRP78 and CHOP activation, SiNP exposure also increased the expression of ERO1α; meanwhile, overexpression and knockdown of CHOP could regulate ERO1α expression. CHOP overexpression increased ERO1α expression and enhanced SiNP-induced apoptosis, while CHOP knockdown decreased ERO1α expression and inhibited SiNP-induced apoptosis. We speculated that the CHOP-ERO1α signaling pathway might play an important role in SiNPs-induced apoptosis in RAW 264.7 macrophage cells. Besides the proapoptotic roles, CHOP, in combination with ATF4 and ATF5, might play a pro-survival role in maintaining mitochondrial function through activation of the mitochondrial unfolded protein response (UPRmt) [45,46]. Possibly, CHOP underwent stress-specific post-translational modifications or heterodimerization that dictated their particular function [47]. Whether SiNPs can activate UPRmt, and whether CHOP participated in SiNP-induced UPRmt, will be determined in a future study.

Recent studies have shown that activation of ER stress-mitochondria-mediated apoptotic pathways might cooperate in SiNP-induced endothelial apoptosis [10]. SiNP induced ER stress might regulate the BCL2 family member BCL-2, BCL-xL, MCL-1, PUMA, and BAX, and ultimately upregulate cytochrome c, Caspase-9 and Caspase-3, leading to the mitochondria-mediated apoptotic caspase cascade [10,24,48,49]. Consistent with previous studies, we found that SiNPs significantly increased proapoptotic BAX, BAD, and Caspase-3, while it decreased antiapoptotic BCL-2. Furthermore, we confirmed that SiNP-induced ER stress-associated activation of CHOP could regulate BCL-2, BCL-xL, MCL-1, PUMA, BAX, and Caspase-3. CHOP overexpression increased PUMA, BAX, and Caspase-3 and decreased BCL-2 expression, while CHOP knockdown decreased PUMA, BAX, and Caspase-3 expression. The proapoptotic BCL-2 family members, such as PUMA, BAX, and BCL-2 homologous antagonist killer (BAK), cause mitochondrial membrane disruption [50,51], and BCL-2-associated death promoter (BAD) and BH3 interacting-domain death agonist (BID) inhibit pro-survival BCL-2 and BCL-xL [52,53]. ER stress upregulates the expression of proapoptotic BAX and BAD, and activates the caspase cascade [54,55]. We speculated that the CHOP-ERO1α-caspase-dependent apoptotic signaling pathway, at least partly, cooperated in SiNP-induced apoptosis in RAW 264.7 macrophage cells.

## 4. Materials and Methods

### 4.1. Reagents

Silica nanoparticles (SiNPs, 637246) were purchased from Sigma-Aldrich (St. Louis, MO, USA). Dulbecco’s modified Eagle’s medium (DMEM), fetal bovine serum (FBS), and Turbofect transfection reagent were purchased from Thermo Fisher Scientific (Waltham, MA, USA). Annexin V-PE/7-AAD kit, total protein extraction kit, and BCA protein assay kit were purchased from Nanjing Keygen Biotech Co., Ltd. (Nanjing, Jiangsu, China). DAPI (4′, 6-diamidino-2-phenylindole) was purchased from Beyotime Institute of Biotechnology (Shanghai, China). Anti-GRP78 antibody (C50B12), anti-cleaved Caspase-3 antibody (9664), and anti-mouse and anti-rabbit HRP-linked secondary antibodies (7076P2, 7074P2) were purchased from Cell Signaling Technology (Danvers, MA, USA). Anti-ERO1α antibody (H00030001-M01) was purchased from Abnova (Taipei, Taiwan). Anti-BCL-2 antibody (ab182858), anti-BCL-xL antibody (ab32370), anti-MCL-1 antibody (ab32087), anti-PUMA antibody (ab33906), anti-BAX antibody (ab32503), anti-BAD antibody (ab32445), and anti-mouse and anti-rabbit fluorescent secondary antibodies (ab150077, ab150115) were purchased from Abcam Ltd. (Cambridge, MA, USA). Anti-CHOP antibody (sc-7351) was purchased from Santa Cruz Biotechnology Inc. (Dallas, TX, USA). Anti-β-actin antibody (KM9001T) was purchased from Tianjin Sungene Biotechnology Inc. (Tianjin, China).

### 4.2. Characterization of SiNPs

The powder of SiNPs was dispersed in sterilized water (final concentration: 4 mg/mL) and stored at 4 °C until use. The morphology and average size of SiNPs were determined by a transmission electron microscope (TEM; Tecnai 12; Royal Philips, Amsterdam, Holland) using dropping aliquots of SiNP solutions on 400-mesh carbon-coated copper grids (SPI Supplies, West Chester, PA, USA). The zeta potential of SiNPs were examined by a Zetasizer 3000HS (Malvern Instruments Ltd., Malvern, UK). Before experiment use, SiNPs in the stock suspension were dispersed using a sonicator (160 W, 20 kHz, 30 min; Bioruptor UCD-200, Belgium).

### 4.3. Cell Line Culture and Treatment

The RAW 264 murine macrophage cell line was cultured in DMEM containing 10% FBS and 1% antibiotic-antimycotic solution in a humidified atmosphere of 5% CO_2_ in air at 37 °C. The cells were passaged every 2–3 days using 0.25% Tyrisin for 1 min at room temperature. When cells reached 70–80% confluence in a plate, they were treated, collected, and processed for further experiments. Firstly, RAW 264.7 macrophages were exposed to various concentrations (0–200 µg/mL) of SiNPs. At various times (0–24 h) during the treatment, cells were collected and processed for CCK-8, flow cytometry, and western blotting detection. Secondly, RAW 264.7 macrophages were exposed to 0, 50, 100, and 150 µg/mL of SiNPs in the presence or absence of 1 mM 4-phenylbutyrate (4-PBA), an ER stress inhibitor. After exposure for 0–24 h, the cells were collected and processed for CCK-8, flow cytometry (CytoFLEX S, Beckman Coulter, Inc., Brea, CA, USA), and western blotting detection.

### 4.4. Measurement of Cell Viability

The effects of SiNPs on cell viability were determined by a CCK-8 assay (C6005, New Cell and Molecular Biotech Co. Ltd., Suzhou, Jiangsu, China). RAW 264 macrophage cells were seeded into a 96-well plate at a density of 1 × 10^4^ cells/200 µL medium/well and incubated for 24 h for attachment. Cells were then replaced with medium containing 0–200 µg/mL SiNPs and subsequently incubated for 12 and 24 h. Next, the CCK-8 solution was then added, and incubated at 37 °C for 2 h. The absorbance at 450 nm was measured at intervals of 1 h with an ELISA plate reader using a microplate reader (Model 680, Bio-Rad, Hercules, CA, USA).

### 4.5. Cell Apoptosis Assay

RAW 264 macrophage cells were cultured into 60 mm dishes at a density of 5 × 10^5^ cells/dish for 12 and 24 h. The cells were then exposed to 0, 50, 100, and 150 µg/mL SiNPs and subsequently incubated for 12 and 24 h. Next, the cells were collected and quantified with an Annexin V-PE and 7-AAD apoptosis detection kit. Following the manufacturer’s instructions, the cells were trypsinized and collected via centrifugation at 2000 rpm for 5 min. The cells were resuspended in 50 μL binding buffer supplemented with 5 μL 7-AAD for 10 min at room temperature in the dark. Then, 450 μL binding buffer was added, followed by the addition of 1 μL Annexin V-PE for another 10 min. Apoptosis was detected using flow cytometry (CytoFLEX S, Beckman Coulter, Inc., Brea, CA, USA) within 1 h.

### 4.6. Immunofluorescence Staining

RAW 264 macrophage cells were cultured on sterile cover slips placed in 24-well culture plates. The cells were then exposed to 100 µg/mL SiNPs, in the presence or absence of 1 mM 4-PBA, and subsequently incubated for 12 h. The cells were then fixed in paraformaldehyde (PFA) solution (4%, *v/v*) overnight at 4 °C. Following fixation, the cells were permeabilized with 0.5% TritonX-100 for 10 min, blocked with 5% BSA for 1 h, and then exposed to anti-CHOP and anti-GRP78 antibodies (1:250 dilutions) overnight at 4 °C. After washing, the cells were incubated with anti-mouse and anti-rabbit fluorescent secondary antibodies (1:1000 dilutions) at 37 °C for 1 h in the dark and DAPI for 10 min at room temperature. Finally, the cells were examined under a laser scanning confocal microscope (TCS SP8 STED; Wetzlar, Hessen, GER). Fluorescence intensity of the images was analyzed with Leica Application Suite X System (Leica, Wetzlar, Hessen, GER).

### 4.7. Construction of Recombinant CHOP Overexpression and Short Hairpin Interfering RNA (shRNA) Lentivirus Plasmid and Cell Transduction

Primers were designed to amplify the CDS sequence of CHOP and cloned into pCD513B-1 plasmids (SBI, Mountain View, CA, USA) to construct a CHOP overexpression lentiviral vector (Lv-CHOP), and pCD513B-1 was used as a negative control group (Lv-Control). Three short hairpin interfering RNA (shRNAs) sequences of CHOP and one negative control sequence were designed and cloned into pCD513B-U6 plasmids to construct CHOP shRNA lentiviral vectors (shCHOP-1, shCHOP-2, and shCHOP-3) and a negative control shRNA vector (shNC). pCD513B-U6 plasmids were reconstructed at the basis of pCD513B-1 [56]. CHOP lentiviral vectors were transfected into HEK 293T cells with three expression vectors encoding the packaging proteins Gag-Pol, Rev, Tat, and the G-protein of the vesicular stomatitis virus (VSVG) using Turbofect transfection reagent, respectively. After transfection for 72 h, the lentivirus-containing supernatants were harvested, purified by low-speed centrifugation, filtered through a 0.45 μm polyvinylidene difluoride (PVDF) filter, and the lentivirus was concentrated using PEG 8000 and stored at −80 °C until use. Lentivirus titer was determined by serial dilution (10^0^–10^7^). RAW 264.7 macrophages were cultured into 60 mm plates at a density of 1 × 10^5^ cells/well. After 24 h, the cells were transduced with an appropriate number of lentiviral particles (multiplicity of infection (MOI) was about 20) in DMEM supplemented with 10% FBS and 8 µg/mL polybrene (Sigma Aldrich, St. Louis, MO, USA). After transduction for 12 h, the lentivirus-containing medium was removed and replaced with fresh DMEM. Following an additional 48 h, the stably expressed cells of CHOP overexpression and shRNAs were selected using puromycin (10 μg/mL, Sigma Aldrich, St. Louis, MO, USA).

### 4.8. Western Blotting

Proteins were extracted from RAW 264 macrophage cells using lysis buffer (Nanjing KeyGen Biotech, Nanjing, China). Protein quantification was measured with a bicinchoninic acid (BCA) protein assay kit (Nanjing KeyGen Biotech, Nanjing, China). Equal amounts of protein (20 µg) were separated by 10% sodium dodecyl sulfate-polyacrylamide gel electrophoresis (SDS–PAGE) and electrotransferred to 0.22-μm PVDF membranes (Millipore, Bedford, MA, USA). Following transfer, membranes were blocked in TBST supplemented with 10% skim milk for 1 h at room temperature and incubated overnight at 4 °C with the relevant primary antibodies. After 12 h, the membranes were incubated with the appropriate HRP-labeled secondary antibodies (1:5000 dilutions) for 1 h at room temperature. Immunoreactive bands were visualized using enhanced chemiluminescence (ECL) reagent (New Cell and Molecular Biotech Co. Ltd., Suzhou, Jiangsu, China) under a Gel Imaging System (Tannon Science and Technology, Shanghai, China), and protein levels were digitized with the Quantity One software (Bio-Rad, Hercules, CA, USA).

### 4.9. Statistical Analysis

Data from the present study are presented as mean ± SDM from at least triplicate independent experiments. Data were analyzed with one-way analysis of variance (ANOVA), followed by Fisher’s least significant different test (Fisher LSD) and an independent samples *t*-test with the Statistical Package for the Social Sciences (SPSS) software (Version 18.0; SPSS, Chicago, IL, USA). The critical value for statistical significance was *p* < 0.05.

## 5. Conclusions

In summary, our results showed that SiNP exposure reduced cell viability and induced cell apoptosis in RAW 264.7 macrophage cells in dose- and time-dependent manners. In parallel, SiNPs induced ER stress by increasing the levels of GRP78, CHOP, and ERO1α expression in RAW 264.7 macrophage cells. The 4-PBA attenuated SiNP-induced ER stress, and inhibited SiNP-induced apoptosis. Overexpression of CHOP increased SiNP-induced apoptosis, while knockdown of CHOP decreased SiNP-induced apoptosis. Regarding the mechanisms, CHOP enhanced the CHOP-mediated and caspase-dependent apoptotic signaling pathways by decreasing the ratio of BCL-2/BAX, BCL-xL, and MCL-1, and increasing downstream ERO1α, PUMA, and Caspase-3 expression. Taken together, we speculated that SiNPs induced ER stress and promoted cell apoptosis, at least partly, via the CHOP-ERO1α-caspase apoptotic signaling pathway. Our findings may provide an in vitro evidence for the potential toxicity of SiNPs, and also offer essential information for a further understanding of toxic mechanisms of SiNPs.

## Figures and Tables

**Figure 1 ijms-20-05846-f001:**
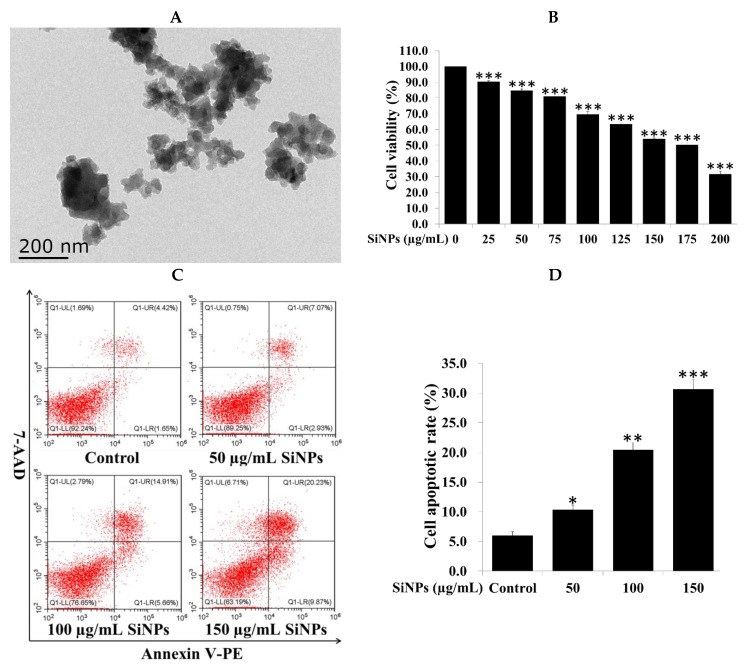
Silica nanoparticles (SiNPs) decreased cell viability and induced cell apoptosis in RAW 264.7 macrophage cells. (**A**) Representative TEM image of SiNPs used in the current work. Scale bar, 200 nm. (**B**) SiNPs reduced cell viability of RAW 264.7 macrophage cells in a dose-dependent manner. Cells were treated with 0, 25, 50, 75, 100, 125, 150, and 200 µg/mL SiNPs for 24 h, and then processed via the CCK-8 assay. (**C**,**D**) Cell apoptosis was detected via flow cytometry analysis. After exposure to 0, 50, 100, and 150 µg/mL SiNPs for 24 h, RAW 264.7 macrophage cells were collected for Annexin V-PE/7-AAD staining. UL quadrant is the part of cell death caused by mechanical damage or necrotic cells, UR quadrant is the part of late apoptotic cells, LL quadrant is the part of the normal cells, and LR quadrant is the part of early apoptotic cells. The number of cell apoptosis included the part of LR quadrant (early apoptotic cells) and UR quadrant (late apoptotic cells). The statistical analysis is shown in the bar graph. Data are presented as the mean ± SDM of three independent experiments. Statistically different from the control is marked with asterisks (* *p* < 0.05, ** *p* < 0.01, and *** *p* < 0.001).

**Figure 2 ijms-20-05846-f002:**
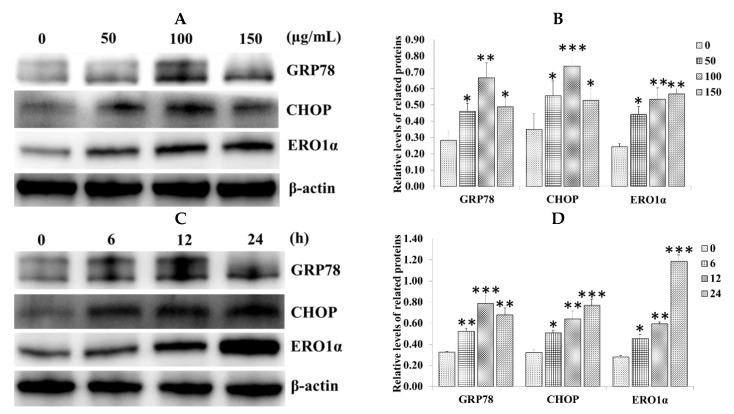
SiNPs induced the expression of the endoplasmic reticulum (ER) stress-related proteins in RAW 264.7 macrophage cells. (**A**,**B**) The expression of glucose-regulated protein 78 (GRP78), CCAAT/enhancer binding protein homologous protein (CHOP), and ER oxidoreduclin 1α (ERO1α) was analyzed via western blot analysis. Cells were exposed to different concentrations of SiNPs (0, 50, 100, and 150 µg/mL) for 12 h; (**C**,**D**) Cells were exposed to different times (0, 6, 12, and 24 h) with 100 µg/mL SiNPs. Analyses of the band intensity on the films are presented as the relative ratio of the related proteins to β-actin. Statistical analysis is shown in the bar graphs. Data are presented as the mean ± SDM of three independent experiments. Statistically different from the control is marked with asterisks (* *p* < 0.05, ** *p* < 0.01, and *** *p* < 0.001).

**Figure 3 ijms-20-05846-f003:**
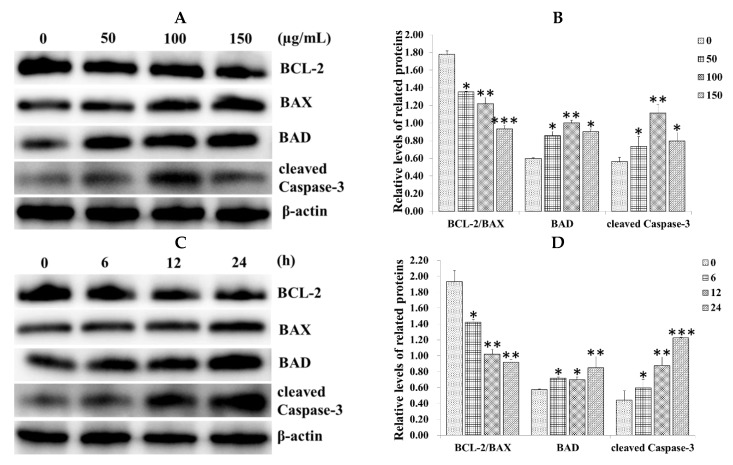
SiNPs induced the expression of the apoptosis-related proteins in RAW 264.7 macrophage cells. (**A**,**B**) The expression of B-cell lymphoma 2 (BCL-2), BCL-2-associated X protein (BAX), BCL-2-associated death promoter (BAD), and cleaved Caspase-3 was analyzed via western blot analysis. Cells were exposed to different concentrations of SiNPs (0, 50, 100, and 150 µg/mL) for 12 h. (**C**,**D**) Cells were exposed to different times (0, 6, 12, and 24 h) with 100 µg/mL SiNPs. Analyses of the band intensity on the films are presented as the relative ratio of the related proteins to β-actin. Statistical analysis is shown in the bar graphs. Data are presented as the mean ± SDM of three independent experiments. Statistically different from the control is marked with asterisks (* *p* < 0.05, ** *p* < 0.01, and *** *p* < 0.001).

**Figure 4 ijms-20-05846-f004:**
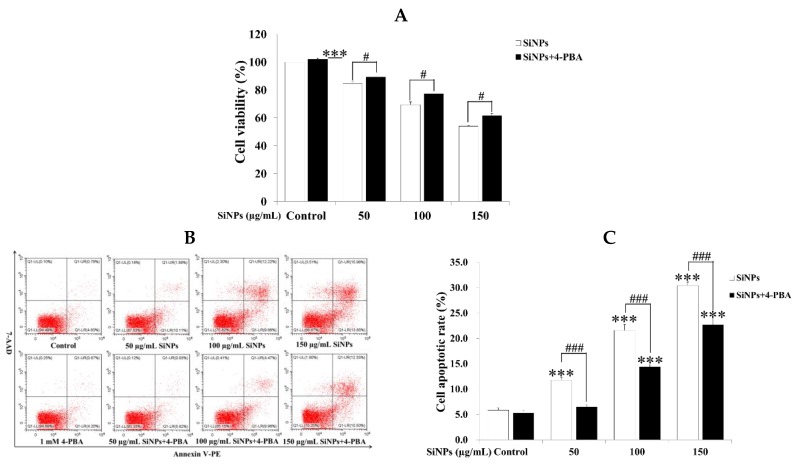
Effect of 4-PBA on cell viability in SiNP-exposed RAW 264.7 macrophage cells. (**A**) Cell viability was determined for the CCK-8 assay. RAW 264.7 macrophage cells were exposed to 0, 50, 100, and 150 µg/mL SiNPs in the presence or absence of 1 mM 4-PBA for 24 h. (**B**,**C**) Cell apoptotic rate was detected via flow cytometry analysis. RAW 264.7 macrophages were pretreated with 1 mM 4-PBA for 1 h, and exposed to 0, 50, 100, and 150 µg/mL SiNPs for 24 h, then collected for Annexin V-PE/7-AAD staining. UL quadrant is the part of cell death caused by mechanical damage or necrotic cells, UR quadrant is the part of late apoptotic cells, LL quadrant is the part of the normal cells, and LR quadrant is the part of early apoptotic cells. The number of cell apoptosis included the part of LR quadrant (early apoptotic cells) and UR quadrant (late apoptotic cells). The statistical analysis is shown in the bar graph. Data are presented as the mean ± SDM of three independent experiments. Statistically different from the control is marked with asterisks (*** *p* < 0.001), and statistically different from SiNPs is marked with number sign (# *p* < 0.05 and ### *p* < 0.001).

**Figure 5 ijms-20-05846-f005:**
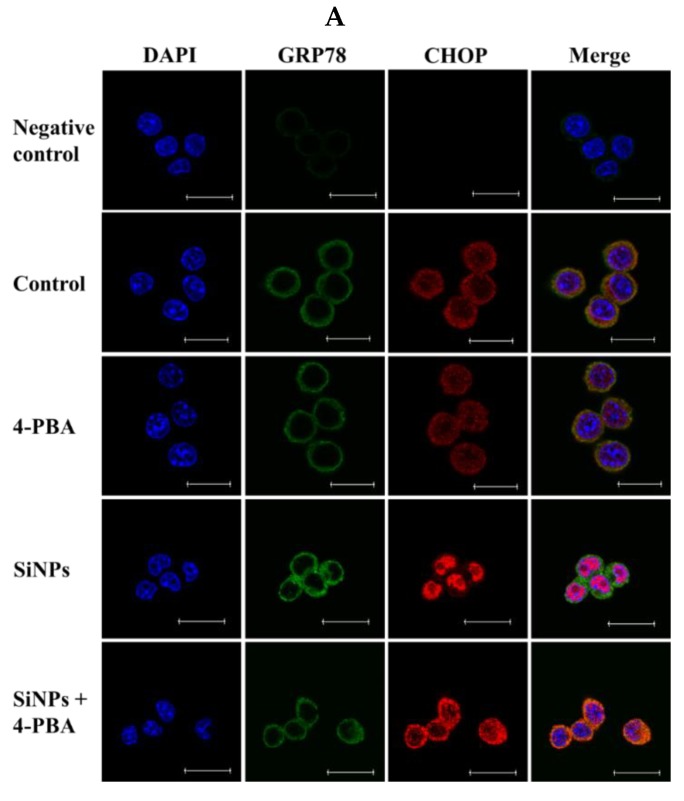
Effect of 4-PBA on the expression of the ER stress- and apoptosis-related proteins in SiNP-exposed RAW 264.7 macrophage cells. (**A**) Confocal immunofluorescence photomicrography showed the expression of GRP78 (green fluorescence) and CHOP (red fluorescence) in the control (the upper panels), SiNP (the center panels) and SiNP + 4-PBA cells (the bottom panels). RAW 264.7 macrophages were pretreated with 1 mM 4-PBA for 1 h, and exposed to 100 µg/mL SiNPs for 12 h. Scale bars, 20 μm. (**B**,**C**) Western blot analysis of GRP78, CHOP, ERO1α, BCL-2, BAX, and cleaved Caspase-3 in SiNP-exposed RAW 264.7 macrophage cells for 12 h. The analyses of the band intensities on films are presented as the relative ratio of target proteins to β-actin. Statistical analysis is shown in the bar graphs. Data are presented as the mean ± SDM of three independent experiments. Statistically different from the control is marked with asterisks (* *p* < 0.05, ** *p* < 0.01, and *** *p* < 0.001), and statistically different from SiNPs is marked with number sign (# *p* < 0.05, ## *p* < 0.01, and ### *p* < 0.001).

**Figure 6 ijms-20-05846-f006:**
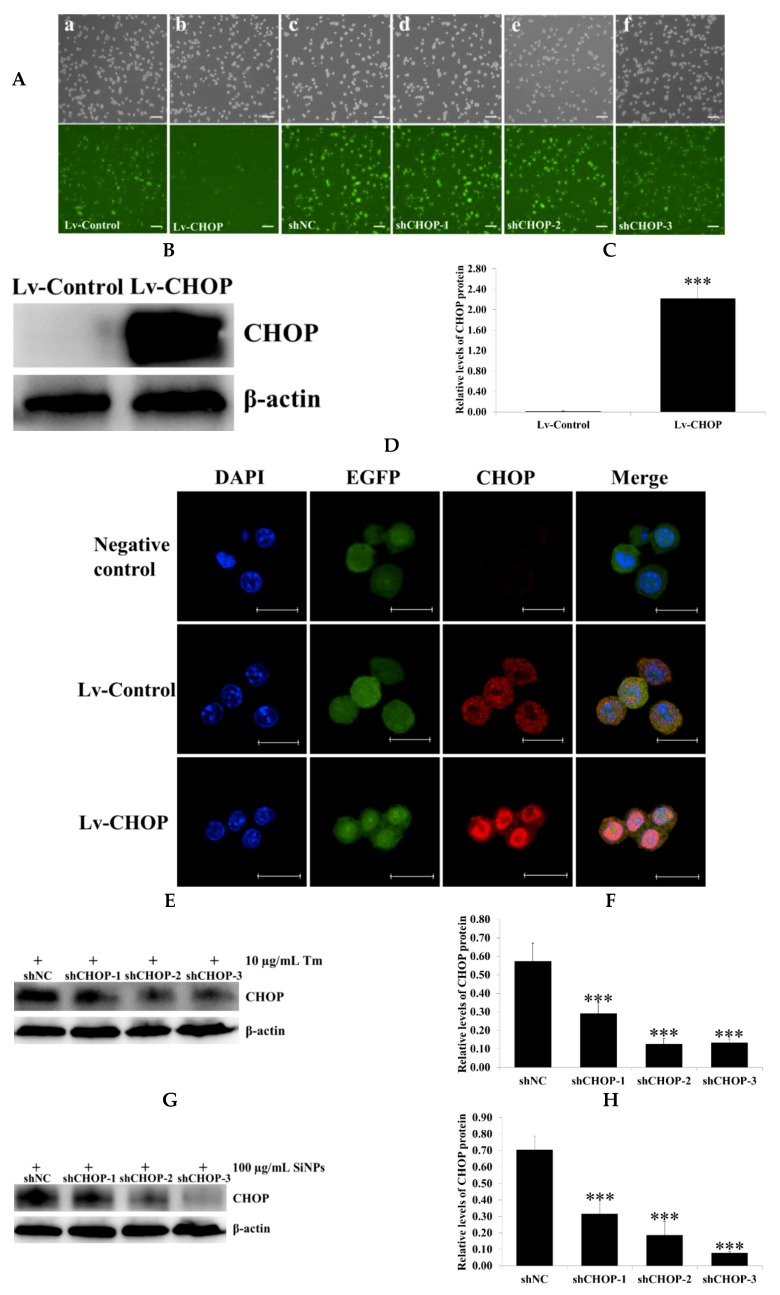
Overexpression and knockdown efficiency of CHOP lentivirus after transduction into RAW 264.7 macrophage cells. RAW 264.7 macrophage cells were transduced with lentivirus vectors (Lv)-CHOP, Lv-Control, three shCHOPs, and shNC lentiviruses, and the stably expressed cells were selected. (**A**) The plasmids themselves could express enhanced green fluorescent protein (EGFP, green fluorescence), and the proportion of the stably expressed cells expressing EGFP was above 95% by flow cytometry. Fluorescence images of the stably expressed cells of the Lv-CHOP (**a**), Lv-Control (**b**), shCHOP-1 (**c**), shCHOP-2 (**d**), shCHOP-3 (**e**), and shNC (**f**) groups; EGFP expression was observed under light (top panels) or fluorescence microscopy (bottom panels). Scale bars, 100 μm. (**B**,**C**) The expression of CHOP was determined in the stably expressed cells of the Lv-CHOP and Lv-Control groups via western blot analysis. (**D**) Immunofluorescence analysis of CHOP (red fluorescence) expression and localization in the stably expressed cells of the Lv-CHOP and Lv-Control groups. Scale bars, 20 μm. (**E**,**F**) The expression of CHOP was determined in the stably expressed cells via western blot analysis. The stably expressed cells of shCHOPs were exposed to 10 μg/mL Tm for 12 h to detect knockdown efficiency. (**G**,**H**) The stably expressed cells of shCHOPs were exposed to 100 μg/mL SiNPs for 12 h to detect knockdown efficiency. The analyses of the band intensity on the films are presented as the relative ratio of CHOP to β-actin. Statistical analysis is shown in the bar graphs. Data are presented as the mean ± SDM of three independent experiments. Statistically different from the control is marked with asterisks (*** *p* < 0.001).

**Figure 7 ijms-20-05846-f007:**
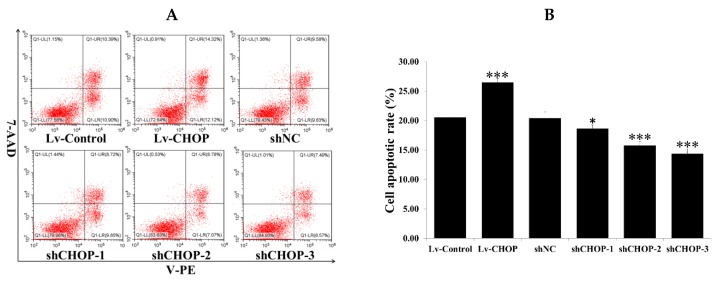
Effect of CHOP on cell apoptosis in RAW 264.7 macrophage cells. (**A**,**B**) The stably expressed cells were exposed to 100 µg/mL SiNPs for 24 h. Cells were stained with Annexin V-PE/7-AAD, and the apoptotic rates were determined via flow cytometry analysis. UL quadrant is the part of cell death caused by mechanical damage or necrotic cells, UR quadrant is the part of late apoptotic cells, LL quadrant is the part of the normal cells, and LR quadrant is the part of early apoptotic cells. The number of cell apoptosis included the part of LR quadrant (early apoptotic cells) and UR quadrant (late apoptotic cells). The statistical analysis is shown in the bar graph. Data are presented as the mean ± SDM of three independent experiments. Statistically different from the control is marked with asterisks (* *p* < 0.05 and *** *p* < 0.001).

**Figure 8 ijms-20-05846-f008:**
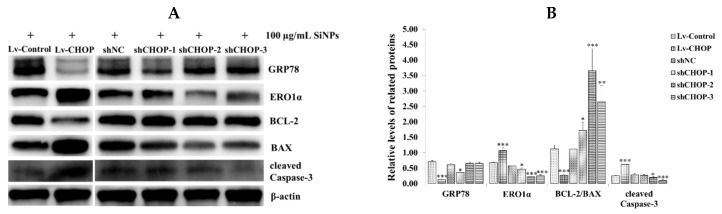
Effect of CHOP on the expression of the ER stress and apoptosis-related proteins in RAW 264.7 macrophage cells. (**A**,**B**) The stably expressed RAW 264.7 macrophage cells were exposed to 100 µg/mL SiNPs for 12 h. The expression of GRP78, CHOP, ERO1α, BCL-2, BAX, and cleaved Caspase-3 were detected via western blot analysis. The analyses of the band intensity on the films are presented as the relative ratio of target proteins to β-actin, respectively. The statistical analysis is shown in the bar graph. Data are presented as the mean ± SDM of three independent experiments. Statistically different from the control is marked with asterisks (* *p* < 0.05, ** *p* < 0.01, and *** *p* < 0.001).

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
