# Peer review of "Endoplasmic Reticulum Stress Cooperates in Silica Nanoparticles-Induced Macrophage Apoptosis via Activation of CHOP-Mediated Apoptotic Signaling Pathway"

_ijms, 2019, doi:10.3390/ijms20235846_

Round 1
Reviewer 1 Report
The manuscript by Lee et al investigates the role of ER stress and the mitochondrial apoptosis pathway in the induction of apoptosis by silica nanoparticles (SiNPs). The authors report that SiNPs induce ER stress in Raw macrophages, resulting in mitochondrial apoptosis induction via upregulation of CHOP and changes of expression of the Bcl-2 family members Bad, Bax and Bcl-2. These are potentially interesting findings which may contribute to a better understanding of the response of macrophages to nanoparticles. However, the conclusions regarding the mechanisms of mitochondrial apoptosis are not completely supported by the data. The authors suggest a causal role for Bad, Bax and Bcl-2, but describe only a correlation. The potential role of further proapoptotic BH3 proteins besides Bad as well as antiapoptotic proteins besides Bcl-2 is ignored and should be investigated.
Major comments:
-Bad is the only BH3 protein which has been investigated. The authors claim that SiNPs activate the mitochondrial apoptotic signaling pathway by upregulation of Bad. However, although the data presented show a correlation between upregulation of Bad and apoptosis induction, this is not sufficient to prove a causal relationship. Such a claim has to be verified by knockdown (e.g. by lentiviral shRNA as they did successfully for testing the role of CHOP) or knockout approaches.
-Other BH3 proteins are possibly involved in mitochondrial apoptosis induction. Bim for instance has been reported to be induced and activated during ER stress. Bim, Puma and Noxa are in many settings more relevant than Bad and should be considered in the current setting of ER stress as well. At least for Bim and Puma, working antibodies are available which allow detection at the endogenous level in many murine cell types. The expression of Bim and Puma in addition to Bad should be analysed by Western blot during SiNP-mediated apoptosis in both the wt and CHOP-knockdown situation. The role of Bim and Puma in SiNP-induced apoptosis should further be tested by knockdown approaches.
-Besides Bcl-2, the anti-apoptotic proteins Bcl-xL and Mcl-1 are important regulators of mitochondrial apoptosis and could be modulated during ER stress-mediated apoptosis, playing a role herein. Expression levels of Bcl-xL and Mcl-1 should be checked.
-Could other forms of apoptosis or cell death play a role as well? Does overexpression of anti-apoptotic Bcl-2 family proteins efficiently protect against SiNP-induced cell death?
Minor comments:
-Page 9, line 230 and 232: The figure numbering of Fig. 6B,C,D in the text has been switched between immunofluorescence and Western blot. Line 230: it should read Fig. 6D (instead of 6B). Line 232: it should read 6B/C (instead of 6C/D).
-The English language needs a little improvement regarding grammar.
Author Response
10-November-2019
International Journal of Molecular Sciences
Manuscript ID: ijms-630020
Title: Endoplasmic reticulum stress cooperates in silica nanoparticles-induced macrophage apoptosis via activation of CHOP-mediated apoptotic signaling pathway
Author: Fenglei Chen*, Jiaqi Jin, Jiahui Hu, Yujing Wang, Zhiyu Ma, Jinlong Zhang
Dear Professor,
On behalf of the authors, we would like to thank you for considering our manuscript “Endoplasmic reticulum stress cooperates in silica nanoparticles-induced macrophage apoptosis via activation of CHOP-mediated apoptotic signaling pathway” for publication in International Journal of Molecular Sciences. We are grateful to you for the comments that can greatly improve our manuscript. We have read the comments carefully and do appreciate for those kind suggestions as well as those critical corrections. Following your suggestions, we have undertaken a revision of the manuscript. Together with the revised manuscript, we include a detailed point-by-point response to your comments. Please feel free to let us know if you have any further questions. Thanks again.
Best Regards,
Fenglei Chen, PhD
College of Veterinary Medicine, Yangzhou University, 12 Wenhui East Road, Yangzhou, 225009, Jiangsu, China;
Tel: +86-514-8797-9030;
Fax: +86-514-8797-2218;
E-mail: flchen@yzu.edu.cn.
Reviewer 1:
Comments:
The manuscript by Lee et al investigates the role of ER stress and the mitochondrial apoptosis pathway in the induction of apoptosis by silica nanoparticles (SiNPs). The authors report that SiNPs induce ER stress in Raw macrophages, resulting in mitochondrial apoptosis induction via upregulation of CHOP and changes of expression of the Bcl-2 family members Bad, Bax and Bcl-2. These are potentially interesting findings which may contribute to a better understanding of the response of macrophages to nanoparticles. However, the conclusions regarding the mechanisms of mitochondrial apoptosis are not completely supported by the data. The authors suggest a causal role for Bad, Bax and Bcl-2, but describe only a correlation. The potential role of further proapoptotic BH3 proteins besides Bad as well as antiapoptotic proteins besides Bcl-2 is ignored and should be investigated.
Response: Thank you for your suggestions and comments. In this study, we focused on the role of ER stress in SiNP-induced apoptosis, and detected whether the CHOP-mediated apoptotic signaling pathway cooperated in ER stress-induced apoptosis. The mitochondrial apoptotic signaling pathway was not study deeply. So, besides Bcl-2 and Bad, the potential roles of other pro-apoptotic and anti-apoptotic BH3 proteins were not measured. Following your suggestions, we further detected the protein expression of BCL-xL, MCL-1, and PUMA in both CHOP-overexpression and CHOP-knockdown situation (Section of Supplementary Materials, Figure S5). This is a good idea for determining the roles and molecular mechanisms of the mitochondrial apoptotic signaling pathway in SiNP-induced apoptosis. We will focus on these studies in the future.
Major comments:
-Bad is the only BH3 protein which has been investigated. The authors claim that SiNPs activate the mitochondrial apoptotic signaling pathway by upregulation of Bad. However, although the data presented show a correlation between upregulation of Bad and apoptosis induction, this is not sufficient to prove a causal relationship. Such a claim has to be verified by knockdown (e.g. by lentiviral shRNA as they did successfully for testing the role of CHOP) or knockout approaches.
Response: Thank you for your suggestion and reminders. We are very sorry that the statement in our conclusion is far too strong; simply based on recent results and we have tone down its significance. Although many studies have showed that the mitochondrial apoptotic signaling pathway participated in SiNPs-induced apoptosis, there is a not sufficient evidence to prove a causal relationship between the mitochondrial apoptotic signaling pathway and SiNP-induced apoptosis. For example, Guo et el. found that SiNPs induced ER stress-associated activation of CHOP, caspase-12, and IRE1a/JNK pathways, which may regulate the BCL2 family member as evidenced by an increased pro-apoptotic BAX while a decline of anti- apoptotic Bcl-2, ultimately facilitate the mitochondria-mediated apoptotic caspase cascade as confirmed by the upregulated expressions of cytochromec, Caspase-9 and -3 [1]. Liu et el. found that SiNPs can result in ovarian damage; cause an imbalance of sex hormones, increase the number of atretic and primary follicles, and induce oxidative stress and DNA strand breaks in ovary, in which process BAX and Caspase-3 were significantly increased. The mitochondrial apoptosis pathway may be activated, resulting in apoptosis of granulosa cells [2]. They found that the ratio of BCL2/BAX was decreased and Caspase-3 was increased in SiNP-induced apoptosis. Consistent with the previous studies, we found that the ratio of BCL-2/BAX, BCL-xL, and MCL-1 was decreased, and PUMA and Caspase-3 was increased. So we speculated that the mitochondrial apoptotic signaling pathway participated in SiNP-induced apoptosis. As you know, if we want to confirm the effect of the mitochondrial apoptotic signaling pathway on SiNP-induced apoptosis, knockdown or knockout approaches will be preceded. This is a good idea and we will focus on the approach to confirm the role of the mitochondrial apoptotic signaling pathway in SiNP-induced apoptosis. We have changed the description to “Taken together, we speculated that SiNPs induced ER stress and promoted cell apoptosis, at least partly, via the CHOP-ERO1α-caspase apoptotic signaling pathway.” in P15, line 470-471 of the manuscript (Section of 4. Conclusion).
-Other BH3 proteins are possibly involved in mitochondrial apoptosis induction. Bim for instance has been reported to be induced and activated during ER stress. Bim, Puma and Noxa are in many settings more relevant than Bad and should be considered in the current setting of ER stress as well. At least for Bim and Puma, working antibodies are available which allow detection at the endogenous level in many murine cell types. The expression of Bim and Puma in addition to Bad should be analysed by Western blot during SiNP-mediated apoptosis in both the wt and CHOP-knockdown situation. The role of Bim and Puma in SiNP-induced apoptosis should further be tested by knockdown approaches.
Response: Thank you for this suggestion. Following your suggestions, we further detected the protein expression of PUMA in both CHOP-overexpression and CHOP-knockdown situation (Section of Supplementary Materials, Figure S5). However, whether BH3 proteins, such as BCL-2, BIM, and PUMA, participate in SiNP-induced apoptosis will be determined and confirmed via knockdown approaches in a future study.
-Besides Bcl-2, the anti-apoptotic proteins Bcl-xL and Mcl-1 are important regulators of mitochondrial apoptosis and could be modulated during ER stress-mediated apoptosis, playing a role herein. Expression levels of Bcl-xL and Mcl-1 should be checked.
Response: We thank you for this suggestion and agree that such data would strengthen the demonstration of the proposed concept. Following your suggestion, the expression levels of BCL-xL and MCL-1 have been checked in both CHOP-overexpression and CHOP-knockdown situation (Section of Supplementary Materials, Figure S5).
-Could other forms of apoptosis or cell death play a role as well? Does overexpression of anti-apoptotic Bcl-2 family proteins efficiently protect against SiNP-induced cell death?
Response: Thank you for your suggestions and reminders. According to the previous researches, autophagy protects against SiNPs-induced toxicity and cell death [3-11]. Besides apoptosis and autophagy, pyroptosis also participates in SiNPs-induced toxicity [12-13].
According to the previous and our studies, the ratio of BCL2/BAX, BCL-xL and MCL-1was decreased, and PUMA and Caspase-3 was increased in SiNP-induced apoptosis. We speculated that overexpression of anti-apoptotic BCL-2 family proteins might efficiently protect against SiNP-induced cell death. This is a good approach and we will focus on it in a future study.
Minor comments:
-Page 9, line 230 and 232: The figure numbering of Fig. 6 B, C, D in the text has been switched between immunofluorescence and Western blot. Line 230: it should read Fig. 6D (instead of 6B). Line 232: it should read 6B/C (instead of 6C/D).
Response: we are very sorry for our mistakes. We have checked the manuscript thoroughly and revised these mistakes you mentioned and others in the manuscript. We have changed the description to “The results of western blot showed that the expression of CHOP in the Lv-CHOP group was significantly increased compared with that in the Lv-Control group (Figure 6B, C). Meanwhile, the results of immunofluorescence staining also showed that CHOP was primarily located in the nucleus of RAW 264.7 macrophage cells and the fluorescence intensity of CHOP in the Lv-CHOP group was stronger than that in the Lv-Control group (Figure 6D).” in Page 8, line 210 to 215 of the manuscript (Section of 2. 5. Verification of the overexpression and knockdown efficiency of recombinant CHOP lentivirus vectors in RAW 264.7 macrophage cells in 2. Results).
-The English language needs a little improvement regarding grammar.
Response: Thank you for your comment, and follow your suggestion, we have edited this manuscript by MDPI Language Editing system, and the English editing ID was english-13778. The certificate wad attached below.
Guo, C. X.; Ma, R.; Liu, X. Y.; Xia, Y. Y.; Niu, P. Y.; Ma, J. X.; Zhou, X. Q.; Li, Y. B.; Sun, Z. W., Silica nanoparticles induced endothelial apoptosis via endoplasmic reticulum stress-mitochondrial apoptotic signaling pathway. Chemosphere 2018, 210, 183-192. Liu, J.; Yang, M.; Jing, L.; Ren, L.; Wei, J.; Zhang, J.; Zhang, F.; Duan, J.; Zhou, X.; Sun, Z., Silica nanoparticle exposure inducing granulosa cell apoptosis and follicular atresia in female Balb/c mice. Environmental science and pollution research international 2018, 25, (4), 3423-3434. Duan, J.; Yu, Y.; Yu, Y.; Li, Y.; Wang, J.; Geng, W.; Jiang, L.; Li, Q.; Zhou, X.; Sun, Z., Silica nanoparticles induce autophagy and endothelial dysfunction via the PI3K/Akt/mTOR signaling pathway. International journal of nanomedicine 2014, 9, 5131-41. Ha, S. W.; Weitzmann, M. N.; Beck, G. R., Jr., Bioactive silica nanoparticles promote osteoblast differentiation through stimulation of autophagy and direct association with LC3 and p62. ACS nano 2014, 8, (6), 5898-910. Yu, Y.; Duan, J.; Yu, Y.; Li, Y.; Liu, X.; Zhou, X.; Ho, K. F.; Tian, L.; Sun, Z., Silica nanoparticles induce autophagy and autophagic cell death in HepG2 cells triggered by reactive oxygen species. Journal of hazardous materials 2014, 270, 176-86. Petrache Voicu, S. N.; Dinu, D.; Sima, C.; Hermenean, A.; Ardelean, A.; Codrici, E.; Stan, M. S.; Zarnescu, O.; Dinischiotu, A., Silica Nanoparticles Induce Oxidative Stress and Autophagy but Not Apoptosis in the MRC-5 Cell Line. Int J Mol Sci 2015, 16, (12), 29398-416. Guo, C.; Yang, M.; Jing, L.; Wang, J.; Yu, Y.; Li, Y.; Duan, J.; Zhou, X.; Li, Y.; Sun, Z., Amorphous silica nanoparticles trigger vascular endothelial cell injury through apoptosis and autophagy via reactive oxygen species-mediated MAPK/Bcl-2 and PI3K/Akt/mTOR signaling. International journal of nanomedicine 2016, 11, 5257-5276. Kretowski, R.; Kusaczuk, M.; Naumowicz, M.; Kotynska, J.; Szynaka, B.; Cechowska-Pasko, M., The Effects of Silica Nanoparticles on Apoptosis and Autophagy of Glioblastoma Cell Lines. Nanomaterials 2017, 7, (8). Marquardt, C.; Fritsch-Decker, S.; Al-Rawi, M.; Diabate, S.; Weiss, C., Autophagy induced by silica nanoparticles protects RAW264.7 macrophages from cell death. Toxicology 2017, 379, 40-47. Wang, J.; Yu, Y.; Lu, K.; Yang, M.; Li, Y.; Zhou, X.; Sun, Z., Silica nanoparticles induce autophagy dysfunction via lysosomal impairment and inhibition of autophagosome degradation in hepatocytes. International journal of nanomedicine 2017, 12, 809-825. Ren, L.; Liu, J.; Zhang, J.; Wang, J.; Wei, J.; Li, Y.; Guo, C.; Sun, Z.; Zhou, X., Silica nanoparticles induce spermatocyte cell autophagy through microRNA-494 targeting AKT in GC-2spd cells. Environmental pollution 2019, 255, (Pt 1), 113172. Du, Q.; Ge, D.; Mirshafiee, V.; Chen, C.; Li, M.; Xue, C.; Ma, X.; Sun, B., Assessment of neurotoxicity induced by different-sized Stober silica nanoparticles: induction of pyroptosis in microglia. Nanoscale 2019, 11, (27), 12965-12972. Zhang, X.; Luan, J.; Chen, W.; Fan, J.; Nan, Y.; Wang, Y.; Liang, Y.; Meng, G.; Ju, D., Mesoporous silica nanoparticles induced hepatotoxicity via NLRP3 inflammasome activation and caspase-1-dependent pyroptosis. Nanoscale 2018, 10, (19), 9141-9152.

Reviewer 2 Report
Manuscript ID: ijms-630020
Type of manuscript: Article
Title: Endoplasmic reticulum stress cooperates in silica nanoparticles-induced macrophage apoptosis via activation of CHOP-mediated apoptotic signaling pathway
Authors: Fenglei Chen *, Jiaqi Jin, Jiahui Hu, Yujing Wang, Zhiyu Ma, jinlong zhang Submitted to section: Molecular Toxicology,
Review
Major points
The authors do not state what and why they are using the old test to detect apoptosis using Hoechst 33342 and PI as opposed to just reporting apoptotic and dead cells from the annexin V data. The authors also do not state clearly what is apoptosis from the annexin V data is it UR and LR quadrants? The DNA dye assay should have given similar results to that of the annexin V assay and as goes often wrong with this test I cannot tell what cells are alive (Double Negative) and what is early apoptotic (Ho33342+ve/PI-ve); so this data should be removed from the manuscript. Dead cells can be reported from the annexin V data using the upper quadrants. The experimental data in Fig. 1C and E is the same as that of the supplementary data 1S. However the results are not the same and given the every low SEM reported in Fig1 this does not make sense, surely the SEM’s should be much larger in Fig1. The statistics should be reported with more detail as the authors do not supply a figure key for their statistical analysis so much so the reader cannot follow how significant the data is. Also the authors should be reporting more than just P<0.05as in Fig1D for example why has 100 and 150 data got a different letter to 50; this implies greater significance which should be reported. The statistics in Fig. 1B does make any sense what so ever? 12h data from Figure 2B does should be the same in part D for GRP78 and CHOP but clearly is not, the authors should explain this discrepancy. The same issue happens again with Fig. 3B and D as well as Fig 4A and C for the 12 h data? P6, line 169, cell apoptosis is not prevented but reduced. The Authors do not state what secondary fluorophores are used to label GRP78 and CHOP used in Figure 4. Also the differences of fluorescence intensity should be measured and quoted, it is not enough to say they ‘are markedly reduced’ (p6, line 170) as the reader wants to know by how much. Figure 5C the SEM bars on the BCL-2/BAX with SiNPS (a) and SiNPS+4-PBS overlap yet the texts says they are significantly different p6, line 175, can this be correct? The authors also do not state how after fixing and permaeablising eGFP cells imaged in Figure 6 how the eGFP signal is preserved by such harsh treatment as GFP does not normally survive such treatment. Again the differences of fluorescence intensity should be measured and quoted, it is not enough to say they ‘are significantly increased in Figure 6D’ (p9, line 232) as the reader wants to know by how much. P12 line 299, the authors cannot state that SiNPs significantly inhibit cell proliferation as they did not test for this cell process. P13, line 353-54 the sentence ‘ER dysfunction may affect mitochondrial dysfunction’ as a standalone statement does not make sense.Minor points
P1, line 16 ‘been focussed on disclose’ should read ‘been focussed on to disclose’. P1, line 16-17 ‘However, only a few literatures’ should read ‘However there are few reports in the literature’. P1, line 38 ‘health effect on nanomaterial’ should read ‘health effect of nanomaterial’. P1, lines 41-43 the syntax should be changed to have just one barrier not six. P2, lines 73-84 should not be just a list of findings but what these mean to this study. Sentence p2, line 86-90 does not make sense on line 88 ‘we do not confirm etc’. P3, line 98, the authors do not state in the Methods how the zeta potential is measured or the significance of this finding? Figure A is missing from the legend and the lettering for Figure 1 is then not correct throughout. P3 line 103 should read different not difference. P4, line 118, SDM should read SEM as is quote in the Supplementary figure 1. This is the same for all the figures in the manuscript. P9, line 217 should state this was the 12 h time point. P10, line 241 full stop missing after CHOP. P14, line 360-370, ‘the purchases’ should be removed from this section. P14, line 372 the authors should state the volume or concentration made. P14, line 388, what flow cytometer was used? P15, line 441 the authors should state a time or overnight ‘the following day’ is not a scientific statement.
Author Response
10-November-2019
International Journal of Molecular Sciences
Manuscript ID: ijms-630020
Title: Endoplasmic reticulum stress cooperates in silica nanoparticles-induced macrophage apoptosis via activation of CHOP-mediated apoptotic signaling pathway
Author: Fenglei Chen*, Jiaqi Jin, Jiahui Hu, Yujing Wang, Zhiyu Ma, Jinlong Zhang
Dear Professor,
On behalf of the authors, we would like to thank you for considering our manuscript “Endoplasmic reticulum stress cooperates in silica nanoparticles-induced macrophage apoptosis via activation of CHOP-mediated apoptotic signaling pathway” for publication in International Journal of Molecular Sciences. We are grateful to you for the comments that can greatly improve our manuscript. We have read the comments carefully and do appreciate for those kind suggestions as well as those critical corrections. Following your suggestions, we have undertaken a revision of the manuscript. Together with the revised manuscript, we include a detailed point-by-point response to your comments. Please feel free to let us know if you have any further questions. Thanks again.
Best Regards,
Fenglei Chen, PhD
College of Veterinary Medicine, Yangzhou University, 12 Wenhui East Road, Yangzhou, 225009, Jiangsu, China;
Tel: +86-514-8797-9030;
Fax: +86-514-8797-2218;
E-mail: flchen@yzu.edu.cn.
Reviewer 2:
Comments:
The authors do not state what and why they are using the old test to detect apoptosis using Hoechst 33342 and PI as opposed to just reporting apoptotic and dead cells from the annexin V data. The authors also do not state clearly what is apoptosis from the annexin V data is it UR and LR quadrants? The DNA dye assay should have given similar results to that of the annexin V assay and as goes often wrong with this test I cannot tell what cells are alive (Double Negative) and what is early apoptotic (Ho33342+ve/PI-ve); so this data should be removed from the manuscript. Dead cells can be reported from the annexin V data using the upper quadrants.
Response: Thank you for your comments and reminders. As you known, the method of Hoechst 33342 and PI confirmed the reliability of observed Annexin V data. As per your suggestions, these data of Hoechst 33342 and PI have been removed from the manuscript.
Thank you for your reminders and following your suggestions, we have added the description “UL quadrant is the part of cell death caused by mechanical damage or necrotic cells, UR quadrant is the part of late apoptotic cells, LL quadrant is the part of the normal cells, and LR quadrant is the part of early apoptotic cells” in Figure 1, 4 and 7 of the manuscript.
The experimental data in Fig. 1C and E is the same as that of the supplementary data 1S. However the results are not the same and given the every low SEM reported in Fig1 this does not make sense, surely the SEM’s should be much larger in Fig1. The statistics should be reported with more detail as the authors do not supply a figure key for their statistical analysis so much so the reader cannot follow how significant the data is. Also the authors should be reporting more than just P<0.05as in Fig1D for example why has 100 and 150 data got a different letter to 50; this implies greater significance which should be reported. The statistics in Fig. 1B does make any sense what so ever? 12h data from Figure 2B does should be the same in part D for GRP78 and CHOP but clearly is not, the authors should explain this discrepancy. The same issue happens again with Fig. 3B and D as well as Fig 4A and C for the 12 h data?
Response: Thank you for your comments and reminders. We are very sorry for our mistakes. The time point of the supplementary data in Figure S1 is 12 h, not 24 h, the time point in Figure 1C and E is 24 h. we have changed the description “12 h” instead of “24 h” in the Figure S1 of the manuscript (Section of Supplementary Materials, Figure S1). According to the method of statistical analysis, data from the present study are presented as mean ± SDM from at least triplicate independent experiments in Page 14, line 456 of the manuscript (Section of 4. 9. Statistical analysis in 4. Materials and Methods). We have changed the description “SDM” instead of “SEM” in the Supplementary Figure S1 of the manuscript and reanalyzed the data of all of the figures in Supplementary Materials.
Thank you for your reminders. We are very sorry that the statistics of Figure 1B were not description clearly. We only performed intra-group comparisons, not inter-group comparisons. In order to avoid confusion, we have added the results of 12 h in the Supplementary Figure S1A (Section of Supplementary Materials, Figure S1) and the results of 24 h in Figure 1B (Section of Figure 1).
Immunoreactive bands were visualized using enhanced chemiluminescence (ECL) reagent (New Cell & Molecular Biotech Co. Ltd, Suzhou, Jiangsu, China) under a Gel Imaging System (Tannon Science & Technology, Shanghai, China). There is a little difference in the aspect of the exposure time, so the bands may be different. We thank you for your reminders and will notice these problems in the future.
P6, line 169, cell apoptosis is not prevented but reduced. The Authors do not state what secondary fluorophores are used to label GRP78 and CHOP used in Figure 4. Also the differences of fluorescence intensity should be measured and quoted, it is not enough to say they ‘are markedly reduced’ (p6, line 170) as the reader wants to know by how much.
Response: Thank you for your comments and reminders. Following your suggestions, we have changed the description “reduced” instead of “prevented” in Page 5, line 156 of the manuscript (Section of 2. 4. Effect of 4-PBA on SiNPs-induced apoptosis in RAW 264.7 macrophage cells in 2. Results).
Following your suggestions, we have added the secondary fluorophores that are used to label GRP78 and CHOP used in Figure 4. The description is that “Meanwhile, 4-PBA markedly reduced the immunofluorescence staining of GRP78 (green fluorescence) and CHOP (red fluorescence) after exposure to 100 µg/mL SiNPs for 12 h in RAW 264.7 macrophage cells (Figure 5A).” in Page 5, line 158 of the manuscript (Section of 2. 4. Effect of 4-PBA on SiNPs-induced apoptosis in RAW 264.7 macrophage cells in 2. Results).
Following your suggestions, we have measured and quoted the difference of fluorescence intensity of Figure 5A. Fluorescence intensity of Figure 5A was analyzed with Leica Application Suite X System (Leica, Wetzlar, Hessen, GER). The results of the statistical analysis have added in Figure S2A of the Supplementary Material.
Figure 5C the SEM bars on the BCL-2/BAX with SiNP (a) and SiNP+4-PBA overlap yet the texts says they are significantly different p6, line 175, can this be correct?
Response: Thank you for your reminders. We have checked and reanalyzed the results of Figure 5B and C. we are very sorry for our mistake. We have changed the description to “while the ratio of BCL-2/BAX was not significantly increased compared with the SiNP exposure group (Figure 5B, C)” in Page 5, line 162-163 of the manuscript (Section of 2. 4. Effect of 4-PBA on SiNPs-induced apoptosis in RAW 264.7 macrophage cells in 2. Results).
The authors also do not state how after fixing and permaeablising eGFP cells imaged in Figure 6 how the eGFP signal is preserved by such harsh treatment as GFP does not normally survive such treatment. Again the differences of fluorescence intensity should be measured and quoted, it is not enough to say they ‘are significantly increased in Figure 6D’ (p9, line 232) as the reader wants to know by how much.
Response: Thank you for your comments and reminders. We are very sorry that immunofluorescence staining of EGFP cells was not descripted in the manuscript. EGFP cells were cultured on sterile cover slips placed in 24-well culture plates. The cells were then fixed in paraformaldehyde (PFA) solution (4%, vol/vol) for 30 min at room temperature. Following fixation, the cells were permeabilized with 0.5% TritonX-100 for 5 min and blocked with 5% BSA for 1 h and then exposed to anti-CHOP (1:250 dilutions) overnight at 4°C. After washing, the cells were incubated with anti-mouse red fluorecscent secondary antibodies (1:1000 dilutions) at 37°C for 1 h in the dark and DAPI for 10 min at room temperature. Finally, the cells were examined under a laser scanning confocal microscope (TCS SP8 STED; Wetzlar, Hessen, GER). The fixing and permaeablising time of eGFP cells was shorter than normal RAW 264.7 macrophage cells. Short-time treatment with PFA can preserve strong enough fluorescence intensity according the research of Zhang et el., 2010 (Zhang G. F., Chen X. G., Shuai P. Q., Lin L., Effect of cell fixative on GFP luminous characteristics. Chem. Bioeng. 2010, 27, (12), 38-40.). We have added the description in the Supplementary Material (Section of 2. 1. Immunofluorescence staining of the stably expressed cells of CHOP in 2. Materials and Methods).
Thank you for your suggestions. Following your suggestions, we have measured and quoted the difference of fluorescence intensity in Figure 6D. Fluorescence intensity of Figure 6D was analyzed with Leica Application Suite X System (Leica, Wetzlar, Hessen, GER). The results of the statistical analysis have added in Figure S2B of the Supplementary Material.
P12 line 299, the authors cannot state that SiNPs significantly inhibit cell proliferation as they did not test for this cell process.
Response: Thank you for your comments and reminders. Following your suggestions, we have changed the description “cell viability” instead of “cell proliferation” in Page 11, line 295-296 of the manuscript (Section of 3. Discussion).
P13, line 353-54 the sentence ‘ER dysfunction may affect mitochondrial dysfunction’ as a standalone statement does not make sense.
Response: Thank you for your comments. As per your suggestion, we have deleted the sentence “ER dysfunction may affect mitochondrial dysfunction”.
Minor points
P1, line 16 ‘been focussed on disclose’ should read ‘been focused on to disclose’.
Response: Thank you for your reminders. We have changed the description “been focused on to disclose” instead of “been focussed on disclose” in Page 1, line 16 of the manuscript (Section of Abstract).
P1, line 16-17 ‘However, only a few literatures’ should read ‘However there are few reports in the literature’.
Response: Thank you for your reminders. We have changed the description “However, there are few reports in the literature” instead of “However, only a few literatures” in Page 1, line 16-17 of the manuscript (Section of Abstract).
P1, line 38 ‘health effect on nanomaterial’ should read ‘health effect of nanomaterial’.
Response: Thank you for your reminders. We have changed the description “health effects of nanomaterial” instead of “health effect on nanomaterial” in Page 1, line 38-39 of the manuscript (Section of 1. Introduction).
P1, lines 41-43 the syntax should be changed to have just one barrier not six.
Response: Thank you for your reminders. We have changed the description to “Due to the unique physicochemical properties of nanoparticles (NPs), they can reach various tissues and organs in the whole body, and penetrate the body’s protective barriers, such as the blood brain barrier, to increase the risk of toxicity to humans and animals.” in Page 1, line 39-42 of the manuscript (Section of 1. Introduction).
P2, lines 73-84 should not be just a list of findings but what these mean to this study.
Response: Thank you for your comments. These finding were proved that numerous studied have reported that ER stress cooperated in SiNP-induced apoptosis and listed the possible molecular mechanisms they have demonstrated.
Sentence p2, line 86-90 does not make sense on line 88 ‘we do not confirm etc’.
Response: Thank you for your comments. As per your suggestion, we have deleted the sentence “Although ER stress is activated in SiNP-induced apoptosis in some cells via detecting the ER stress-related proteins, such as GRP78, PERK, IRE1α, ATF6, ATF4, XBP1 or CHOP, we want to confirm that ER stress enhances or inhibits SiNPs-induced apoptosis, and whether CHOP-mediated apoptotic signaling pathway plays an important role.”.
P3, line 98, the authors do not state in the Methods how the zeta potential is measured or the significance of this finding? Figure A is missing from the legend and the lettering for Figure 1 is then not correct throughout.
Response: Thank you for your suggestions and reminders. We have added the description to “Furthermore, to detect the electrical property of SiNPs, the result showed that SiNPs were electronegative in water, and the zeta potential was measured as -9.9 ± 1.1 mV.” in Page 3, line 93-94 of the manuscript (Section of 2.1. SiNPs characterization and cell viability and apoptosis in RAW 264.7 macrophage cells in 2. Results).
Thank you for your reminders. We are very sorry for our mistake. We have added the description to “(A) Representative TEM image of SiNPs used in the current work. Scale bar, 200 nm. (B) SiNPs reduced cell viability of RAW 264.7 macrophage cells in a dose-dependent manner. Cells were treated with 0, 25, 50, 75, 100, 125, 150, and 200 µg/mL SiNPs for 24 h, and then processed via the CCK-8 assay.” in Page 3, line 104-107 of the manuscript (Section of Figure 1 in 2. Results).
P3 line 103 should read different not difference.
Response: Thank you for your reminders. We have changed the description “different” instead of “difference” in Page 3, line 100 of the manuscript (Section of 2.1. SiNPs characterization and cell viability and apoptosis in RAW 264.7 macrophage cells in 2. Results).
P4, line 118, SDM should read SEM as is quote in the Supplementary figure 1. This is the same for all the figures in the manuscript.
Response: Thank you for your reminders. We are very sorry for our mistakes. We have checked the manuscript thoroughly and revised these mistakes you mentioned in the manuscript. According to the method of statistical analysis, data from the present study are presented as mean ± SDM from at least triplicate independent experiments in Page 16, line 456-460 of the manuscript (Section of 4. 9. Statistical analysis in 4. Materials and Methods). We have changed the description “SDM” instead of “SEM” in the all of figures in Supplementary Materials.
P9, line 217 should state this was the 12 h time point.
Response: Thank you for your reminders. We have added the 12 h time point and changed the description to “(B, C) Western blot analysis of GRP78, CHOP, ERO1α, BCL-2, BAX and cleaved Caspase-3 in SiNP-exposed RAW 264.7 macrophage cells for 12 h.” in Page 7, line 199 of the manuscript (Section of Figure 5 in 2. Results).
P10, line 241 full stop missing after CHOP.
Response: Thank you for your reminders. We have checked and revised the mistake in Page 9, line 233 of the manuscript (Section of Figure 6 in 2. Results).
P14, line 360-370, ‘the purchases’ should be removed from this section.
Response: Thank you for your suggestions. We think that the reagents should mark the sources; the reagents of different companies may have different effects and protocols. In the method of 4.9 Western blot, because of many primary antibodies, we did not list them in the section of 4. 9. Western blotting in the 4. Materials and Methods, so it is necessary to show them in the section of 4. 1. Reagents in the 4. Materials and Methods of the manuscript.
P14, line 372 the authors should state the volume or concentration made.
Response: Thank you for your reminders. We have added the concentration of the SiNPs solution and changed the description to “The powder of SiNPs were dispersed in sterilized water (Final concentration: 4 mg/mL) and stored at 4°C until use.” in Page 13, line 375 of the manuscript (Section of 4.2. Characterization of SiNPs in 4. Materials and Methods).
P14, line 388, what flow cytometer was used?
Response: Thank you for your reminders. We have added the model of flow cytometer and changed the description to “After exposure for 0–24 h, the cells were collected and processed for CCK-8, flow cytometry (CytoFLEX S, Beckman Coulter, Inc., Brea, CA, USA) and western blotting detection.” in Page 13, line 391-392 of the manuscript (Section of 4. 3. Cell line culture and treatment in 4. Materials and Methods).
P15, line 441 the authors should state a time or overnight ‘the following day’ is not a scientific statement.
Response: Thank you for your reminders. We have changed the description to “After 24 h, the cells were transduced with an appropriate number of lentiviral particles (multiplicity of infection (MOI) was about 20) in DMEM that is supplemented with 10% FBS and 8 µg/mL polybrene (Sigma Aldrich, St. Louis, MO, USA).” in Page 14, line 436-438 of the manuscript (Section of 4. 8. Construction of recombinant CHOP overexpression and shRNAs lentivirus plasmid and cell transduction in 4. Materials and Methods).

Reviewer 3 Report
This is a well executed study to explore role of CHOP in ER stress mediated apoptosis in response to SiNPs treatment. The results were clear and manuscript was well written. The only major concern is the significance of the CHOP in SiNPs mediated apoptosis. From both 4-PBA and genetic manipulations of CHOP results, the ER stress regardless through CHOP or non-CHOP pathways only contributed to 5-10% of apoptosis induced by SiNPs, suggesting ER stress only plays a minor role. In figure 5A, the cells with CHOP nuclear localisation also looked healthy. Authors only focused on the pro-apoptotic role of CHOP, however, recent studies have also revealed the protective side of CHOP in preservation of mitochondrial function through direct activation of mitochondrial unfolded protein response pathway. These may explain the shuttle effect of CHOP in SiNPs mediate apoptosis. Therefore, authors should also mention this potential protective role of CHOP in the discussion.
Specific comments:
Many of western blot results were saturation, results with shorter exposure time should be used and presented. In figure 4, the protective effect of 4-PBA is limited (around 5-10%) regardless the increased concentrations of SiNPs indicating the limited contribution of ER stress in siNPs-induced apoptosis. Did author try a combination of anti-oxidant and 4-PBA together? The may provide a better protective effect especially mitochondrial mediated apoptosis also activated upon SiNPs treatment. In Fig 5A, immunofluorescent images showed a complete inhibition of the nuclear translocation of CHOP in SiNPs + 4-PBA treated cells. Could authors quantify and present the percentage of CHOP nuclear localisation of all 4 conditions? The current images were a bit of misleading. Additionally, could authors also quantify and present the percentage of apoptotic cells with CHOP nuclear localisation in those conditions. Figure 6 should be presented as supplementary figure. Was there any cell death in LvCHOP expressed cells in the absence of SiNPs treatment? Did authors look for ER resided caspase 4 activation. ER stress can induce apoptosis directly through caspase 4 independent of CHOP. Could author explain in figure 8 why in LvCHOP condition, GRP78 was reduced? In figure 8, why changed the incubation time of SiNPs from 24 to 48 hr. However, there was no increased in cell death (~25%) in control (shNC) after 48 hr, similar to 24 hr result in Fig 1A? How could authors explain it? Did authors try to use higher concentration of SiNPs (such as 200 ug/ml which induces 75% apoptosis) in shCHOP study to see whether they have stronger protective effect? Did author measure mitochondrial membrane potential in cells with LvCHOP and shCHOP? This will give clues whether changed in CHOP levels alters mitochondrial function. Either overexpressed CHOP in LvCHOP or knockdown of CHOP only affected SiNPs induced apoptosis by 5-10%, suggesting CHOP only playing a minor role. Unless there are stronger results to support it, otherwise, the statement in discussion in line 297-298 is far too strong, simply based on 5-10% changes and it will need to tone down its significance.Author Response
10-November-2019
International Journal of Molecular Sciences
Manuscript ID: ijms-630020
Title: Endoplasmic reticulum stress cooperates in silica nanoparticles-induced macrophage apoptosis via activation of CHOP-mediated apoptotic signaling pathway
Author: Fenglei Chen*, Jiaqi Jin, Jiahui Hu, Yujing Wang, Zhiyu Ma, Jinlong Zhang
Dear Professor,
On behalf of the authors, we would like to thank you for considering our manuscript “Endoplasmic reticulum stress cooperates in silica nanoparticles-induced macrophage apoptosis via activation of CHOP-mediated apoptotic signaling pathway” for publication in International Journal of Molecular Sciences. We are grateful to you for the comments that can greatly improve our manuscript. We have read the comments carefully and do appreciate for those kind suggestions as well as those critical corrections. Following your suggestions, we have undertaken a revision of the manuscript. Together with the revised manuscript, we include a detailed point-by-point response to your comments. Please feel free to let us know if you have any further questions. Thanks again.
Best Regards,
Fenglei Chen, PhD
College of Veterinary Medicine, Yangzhou University, 12 Wenhui East Road, Yangzhou, 225009, Jiangsu, China;
Tel: +86-514-8797-9030;
Fax: +86-514-8797-2218;
E-mail: flchen@yzu.edu.cn.
Reviewer 3:
Comments:
This is a well executed study to explore role of CHOP in ER stress mediated apoptosis in response to SiNPs treatment. The results were clear and manuscript was well written. The only major concern is the significance of the CHOP in SiNPs mediated apoptosis. From both 4-PBA and genetic manipulations of CHOP results, the ER stress regardless through CHOP or non-CHOP pathways only contributed to 5-10% of apoptosis induced by SiNPs, suggesting ER stress only plays a minor role. In figure 5A, the cells with CHOP nuclear localization also looked healthy. Authors only focused on the pro-apoptotic role of CHOP, however, recent studies have also revealed the protective side of CHOP in preservation of mitochondrial function through direct activation of mitochondrial unfolded protein response pathway. These may explain the shuttle effect of CHOP in SiNPs mediate apoptosis. Therefore, authors should also mention this potential protective role of CHOP in the discussion.
Response: We thank you for your comments and agree that such suggestions would strengthen the demonstration of the proposed concept. In this study, we demonstrated that ER stress cooperated in SiNP-induced apoptosis and CHOP plays an important role in ER stress-mediate apoptosis. There are many signaling pathways activated in SiNP-induced apoptosis, such as oxidative stress, autophagy and mitochondrial-mediated apoptotic signaling pathway. SiNPs induced apoptosis via excessive reactive oxygen species (ROS) generation-mediate oxidative stress at first. Along with the deepening of research, besides oxidative stress, excessive ROS also affects endoplasmic reticulum (ER) homeostasis leading to ER stress. So many researchers focus on the effect of ER stress-mediate apoptosis after SiNP exposure. The ER stress regardless through CHOP or non-CHOP pathways only contributed to 5-10% of apoptosis induced by SiNPs, ER stress may have different effects in different cells. However, as per your suggestions, 200 µg/ml SiNPs were used to detect the effect of CHOP to see whether they have stronger protective effect. The results of flow cytometry showed that the apoptotic rate of the shCHOP-3 group (26.58%) is lower than that of the shNC group (41.20%) (Section of Supplementary Material, Figure S4B). Knockdown of CHOP contributed to about 15% of apoptosis induced by 200 µg/ml SiNPs for 24 h. Along with increasing dose, knockdown of CHOP may have stronger protective effects.
Thank you for your comments. The stably expressed cells of CHOP are healthy without the condition of stress. Following your suggestions, the stably expressed cells were no treatment and stained with Annexin V-PE/7-AAD, and the apoptotic rates were determined via flow cytometry analysis. The results of flow cytometry assays showed that there is no significant difference between the Lv-CHOP group and the Lv-Control group (Section of Figure S4A in the Supplementary Materials).
Thank you for your reminders. Following your suggestions, we have added the description about the protective side of CHOP in preservation of mitochondrial function through direct activation of mitochondrial unfolded protein response pathway in Page 12, line 337-342 of the manuscript (Section of 3. Discussion). The description is “Besides the proapoptotic roles, CHOP, in combination with ATF4 and ATF5, might play a prosurvival role in maintaining mitochondrial function through activation of the mitochondrial unfolded protein response (UPRmt) [46, 47]. Possibly, CHOP underwent stress-specific post-translational modifications or heterodimerization that dictated their particular function [48]. Whether SiNPs can activate UPRmt, and whether CHOP participated in SiNP-induced UPRmt will be determined in a future study.”
Specific comments:
Many of western blot results were saturation, results with shorter exposure time should be used and presented. In figure 4, the protective effect of 4-PBA is limited (around 5-10%) regardless the increased concentrations of SiNPs indicating the limited contribution of ER stress in siNPs-induced apoptosis. Did author try a combination of anti-oxidant and 4-PBA together? The may provide a better protective effect especially mitochondrial mediated apoptosis also activated upon SiNPs treatment.
Response: Thank you for your comments and suggestions. We appreciate you for your reminders and will notice these problems of the saturation of western blot results in the future. This is a good idea that tries a combination of anti-oxidant and 4-PBA together. Maybe combination of anti-oxidant and 4-PBA together will contribute greatly to SiNP-induced apoptosis. Anti-oxidant can effectively inhibit excessive ROS generation and ROS-induced oxidative stress and ER stress, and provide a better protective effect, especially mitochondrial mediated apoptosis. We will proceed with the related study to demonstrate our hypothesis in a future study.
In Fig 5A, immunofluorescent images showed a complete inhibition of the nuclear translocation of CHOP in SiNPs + 4-PBA treated cells. Could authors quantify and present the percentage of CHOP nuclear localisation of all 4 conditions? The current images were a bit of misleading. Additionally, could authors also quantify and present the percentage of apoptotic cells with CHOP nuclear localisation in those conditions.
Response: Thank you for your suggestions and reminders. Following your suggestions, we have measured and quoted the difference of fluorescence intensity of Figure 6D. Fluorescence intensity of Figure 6D was analyzed with Leica Application Suite X System (Leica, Wetzlar, Hessen, GER). The results of the statistical analysis have added in Figure S2A of the Supplementary Material.
Thank you for your comments. The stably expressed cells of CHOP are healthy without the condition of stress. Following your suggestions, the stably expressed cells were no treatment and stained with Annexin V-PE/7-AAD, and the apoptotic rates were determined via flow cytometry analysis. The results of flow cytometry assays showed that there is no significant difference between the Lv-CHOP group and the Lv-Control group (Section of Figure S4A in the Supplementary Materials).
Figure 6 should be presented as supplementary figure. Was there any cell death in LvCHOP expressed cells in the absence of SiNPs treatment? Did authors look for ER resided caspase 4 activation. ER stress can induce apoptosis directly through caspase 4 independent of CHOP.
Response: Thank you for your suggestions. The results of Figure 6 confirmed the overexpression and knockdown efficiency of recombinant CHOP lentivirus vectors in RAW 264.7 macrophage cells. We think that it is relatively important, especially for the following experiment to demonstrate the role of CHOP in ER stress-induced apoptosis after SiNP treatment.
Following your suggestions, the stably expressed cells were no treatment and stained with Annexin V-PE/7-AAD, and the apoptotic rates were determined via flow cytometry analysis. The results of flow cytometry in the LvCHOP expressed cells in the absence of SiNP treatment have added in Figure S4A of the Supplementary Material. The results of flow cytometry assays showed that there is no significant difference between the Lv-CHOP group (6.19%) and the Lv-Control group (5.47%). However, the apoptotic rate of the Lv-CHOP group is higher than that of the shNC group (3.81%) (Section of Supplementary Material, Figure S4A).
Thank you for your reminders. We will focus on proceeding with the related study to demonstrate whether ER resided Caspase 4 was activated after SiNP treatment in a future study.
Could author explain in figure 8 why in LvCHOP condition, GRP78 was reduced? In figure 8, why changed the incubation time of SiNPs from 24 to 48 hr. However, there was no increased in cell death (~25%) in control (shNC) after 48 hr, similar to 24 hr result in Fig 1A? How could authors explain it? Did authors try to use higher concentration of SiNPs (such as 200 ug/ml which induces 75% apoptosis) in shCHOP study to see whether they have stronger protective effect?
Response: Thank you for your comments. The 78-kDa glucose-regulated protein (GRP78), also known as heat shock 70 kDa protein 5 (HSPA5) or immunoglobulin heavy chain-binding proteins (BiP), is a member of the heat shock protein 70 (Hsp70) family. GRP78 is present at multiple subcellular locations and has roles in several cellular processes, including serving as an indicator for the onset of ER stress, facilitating the correct folding and assembly of newly synthesized proteins, preventing aggregation of unfolded peptides, targeting misfolded proteins for proteasomal degradation, maintaining ER calcium homeostasis, recognizing extracellular ligands, and transducing proliferative signals. Furthermore, because of its anti-apoptotic property, GRP78 might be upregulated to protect the host cell against ER stress. GRP78 can protect the cells against a variety of physiological stresses through unfolded protein response (UPR) signaling, which affects complex signal transduction pathways designed to restore ER homeostasis. GRP78 binds toinositol-requiring enzyme 1α (IRE1α), double-stranded RNA-dependent protein kinase (PKR)-like ER kinase(PERK), and activating transcription factor (ATF)-6 in homeostasis, all of which are ER-localized protein sensors for UPR. When these sensors recognize more significant stress, GRP78 separates from these sensors and interacts with misfolded or unfolded proteins to restore homeostasis. However, severe or prolonged ER stress will trigger cell death. Cell death proceeds through obvious pathways including the activation of CHOP, JNK and caspase-12. CHOP, an apoptotic transcriptional factor induced in response to ER stress, is also a popular marker for the assessment of ER stress. So activation of CHOP will inhibit the function of GRP78.
Thank you for your reminders. We are very sorry for our mistakes. The incubation time of SiNPs was for 24 h, not for 48 h. We have changed the description “24 h” instead of “48 h” in Page 10, line 264 of the manuscript (Section of Figure 7).
Thank you for your suggestions. As per your suggestions, 200 µg/ml SiNPs were used to detect the effect in shCHOP study to see whether they have stronger protective effect. The results of flow cytometry showed that the apoptotic rate of the shCHOP-3 group (26.58%) is lower than that of the shNC group (41.20%) (Section of Supplementary Material, Figure S4B). Knockdown of CHOP contributed to about 15% of apoptosis induced by 200 µg/ml SiNPs for 24 h.
Did author measure mitochondrial membrane potential in cells with LvCHOP and shCHOP? This will give clues whether changed in CHOP levels alters mitochondrial function. Either overexpressed CHOP in LvCHOP or knockdown of CHOP only affected SiNPs induced apoptosis by 5-10%, suggesting CHOP only playing a minor role. Unless there are stronger results to support it, otherwise, the statement in discussion in line 297-298 is far too strong, simply based on 5-10% changes and it will need to tone down its significance.
Response: Thank you for your comments and reminders. In this study, we focused on the role of ER stress in SiNP-induced apoptosis, and detected whether CHOP-mediated apoptotic signaling pathway plays an important role in ER stress-induced apoptosis. The mitochondrial apoptotic signaling pathway was not study deeply, of cause, the mitochondrial membrane potential in cells with Lv-CHOP and shCHOP was not measured. This is a good idea for determining the role and molecular mechanism of the mitochondrial apoptotic signaling pathway in SiNP-induced apoptosis. We will focus on these studies in the future.
Thank you for your reminders, and following your suggestions, we have changed the description to “In the current study, we demonstrated that SiNPs induced cell apoptosis, at least partly, via ER stress activation in RAW 264.7 macrophage cells, of which the CHOP-mediated apoptotic signaling pathway might play an important role in ER stress-induced apoptosis.” in Page 11, line 292-294 of the manuscript (Section of 3. Discussion).

Round 2
Reviewer 1 Report
The authors have adequately adressed most of the comments and concerns raised by the reviewers
Author Response
16-November-2019
International Journal of Molecular Sciences
Manuscript ID: ijms-630020
Title: Endoplasmic reticulum stress cooperates in silica nanoparticles-induced macrophage apoptosis via activation of CHOP-mediated apoptotic signaling pathway
Author: Fenglei Chen*, Jiaqi Jin, Jiahui Hu, Yujing Wang, Zhiyu Ma, Jinlong Zhang
Dear Professor,
We would like to thank you for reconsidering our manuscript “Endoplasmic reticulum stress cooperates in silica nanoparticles-induced macrophage apoptosis via activation of CHOP-mediated apoptotic signaling pathway” for publication in International Journal of Molecular Sciences. We are grateful to you for your positive comments. We have checked and revised our manuscript again. Thanks again.
Best Regards,
Fenglei Chen, PhD
College of Veterinary Medicine, Yangzhou University, 12 Wenhui East Road, Yangzhou, 225009, Jiangsu, China;
Tel: +86-514-8797-9030;
Fax: +86-514-8797-2218;
E-mail: flchen@yzu.edu.cn.
Reviewer 1:
Comments:
The authors have adequately adressed most of the comments and concerns raised by the reviewers
Response: Thanks for your affirmation. We have checked and revised our manuscript again.

Reviewer 2 Report
The Authors have made the significant requested or few very minor corrections are requested. The issue however of not saying precisely what the statistics are in each figure SHOULD BE ADDRESSED as REQUESTED PREVIOUSLY, as the currently lettering system without the provision of a KEY is not working for the reader and is very confusing to say the least.

Author Response
16-November-2019
International Journal of Molecular Sciences
Manuscript ID: ijms-630020
Title: Endoplasmic reticulum stress cooperates in silica nanoparticles-induced macrophage apoptosis via activation of CHOP-mediated apoptotic signaling pathway
Author: Fenglei Chen*, Jiaqi Jin, Jiahui Hu, Yujing Wang, Zhiyu Ma, Jinlong Zhang
Dear Professor,
We would like to thank you for reconsidering our manuscript “Endoplasmic reticulum stress cooperates in silica nanoparticles-induced macrophage apoptosis via activation of CHOP-mediated apoptotic signaling pathway” for publication in International Journal of Molecular Sciences. We are grateful to you for your reminders and further explanations that can greatly improve our manuscript. We have read your reminders and further explanations carefully and do appreciate for those kind suggestions as well as those critical corrections. Following your suggestions, we have undertaken a revision of the manuscript. Together with the revised manuscript, we include a detailed point-by-point response to your comments. Please feel free to let us know if you have any further questions. Thanks again.
Best Regards,
Fenglei Chen, PhD
College of Veterinary Medicine, Yangzhou University, 12 Wenhui East Road, Yangzhou, 225009, Jiangsu, China;
Tel: +86-514-8797-9030;
Fax: +86-514-8797-2218;
E-mail: flchen@yzu.edu.cn.
Reviewer 2:
Comments:
The Authors have made the significant requested or few very minor corrections are requested. The issue however of not saying precisely what the statistics are in each figure SHOULD BE ADDRESSED as REQUESTED PREVIOUSLY, as the currently lettering system without the provision of a KEY is not working for the reader and is very confusing to say the least. But what quadrants did you use to calculate ‘Cell apoptotic rate (%)’ this should be made clear to the reader were the upper 2 quadrants used together with the LR or some other combination. I asked before to make it clear what EXACTLY are the STATICS for each data point, YOU PROVIDE NOT KEY TO THE LETTERS, is the letter ‘a’ not significant ‘b’ P<0.05, ‘c’ P<0.01 and ‘d’ P<0.001 - as this is what it looks like but the Authors should state EXACTLY WAHY ARE THE STATICS! As this lettering system is impossible to decipher WITHOUT A KEY to the letters!
Response: Thank you for your comments and reminders. We are very sorry to incompletely understand your meaning before and appreciate your further explanations. Following your suggestions, we have added the description that how to calculate cell apoptotic rate. The revised description is “The number of cell apoptosis included the part of LR quadrant (early apoptotic cells) and UR quadrant (late apoptotic cells).” in Figure1, 4, and 7 of the manuscript and Figure S1 and S4 of the Supplementary Material.
Thank you for your reminders and further explanations. Following your suggestions, we have checked and reanalyzed the statistics, added the description about the statistical significance in all figure legends, and revised all related figures in the manuscript and the Supplementary Material. The revised description is “Statistically different from the control is marked with asterisk (*p < 0.05, **p < 0.01, and ***p < 0.001), and statistically different from SiNPs is marked with number sign (#p < 0.05, ##p < 0.01, and ###p < 0.001).”. According to the guidance of Prof. Huiguang Wu (College of Veterinary Medicine, Yangzhou University), who mainly engages in bioinformatics analysis, he said that it would be better to understand what we meant.
P6, line 169, cell apoptosis is not prevented but reduced. The Authors do not state what secondary
fluorophores are used to label GRP78 and CHOP used in Figure 4. The Authors should state WHAT Fluorophores precisely where used, I can see that they are GREEN & RED but what was used e.g. FITC, AF488, Texas Red or WHAT?
Response: Thank you for your comments and reminders. We have changed the description “inhibited” instead of “reduced” in Page 5, line 156 of the manuscript (Section of 2. 4. Effect of 4-PBA on SiNP-induced apoptosis in RAW 264.7 macrophage cells in 2. Results).
We are very sorry to incompletely understand your meaning before and appreciate your further explanations. Following your suggestions, we have added the fluorophores precisely. The revised description is “GRP78 (green fluorescence, Alexa Fluor® 488) and CHOP (red fluorescence, Alexa Fluor® 647)” in Page 5, Line 158-159 of manuscript (Section of 2. 4. Effect of 4-PBA on SiNP-induced apoptosis in RAW 264.7 macrophage cells in 2. Results).
NEW COMMENTS, this is a great improvement on the first manuscript just a minor issue of using the term ‘inhibited’ cell viability should be changed to read ‘reduced cell viability’, this occurs on P3, line 142, P11, line 482 and P15, line 731
Response: Thank you for your comments and reminders. Following your suggestions, we have changed the description “reduced” instead of “inhibited” in Page 3, line 97 and Page 15, line 460 of the manuscript (Section of 2.1. SiNP characterization and cell viability and apoptosis in RAW 264.7 macrophage cells in 2. Results, and 5. Conclusions).
